# A RAD51–ADP double filament structure unveils the mechanism of filament dynamics in homologous recombination

Shih-Chi Luo [1], Min-Chi Yeh[1,2], Yu-Hsiang Lien[3], Hsin-Yi Yeh[2], Huei-Lun Siao[3], I-Ping Tu [3], Peter Chi [1,2] & Meng-Chiao Ho [1,2] ✉

ATP-dependent RAD51 recombinases play an essential role in eukaryotic homologous recombination by catalyzing a four-step process: 1) formation of a RAD51 single-filament assembly on ssDNA in the presence of ATP, 2) complementary DNA strand-exchange, 3) ATP hydrolysis transforming the RAD51 filament into an ADP-bound disassembly-competent state, and 4) RAD51 disassembly to provide access for DNA repairing enzymes. Of these steps, filament dynamics between the ATP- and ADP-bound states, and the RAD51 disassembly mechanism, are poorly understood due to the lack of near-atomic-resolution information of the ADP-bound RAD51–DNA filament structure. We report the cryo-EM structure of ADP-bound RAD51–DNA filaments at 3.1 Å resolution, revealing a unique RAD51 double-filament that wraps around ssDNA. Structural analysis, supported by ATP-chase and time-resolved cryo-EM experiments, reveals a collapsing mechanism involving two four-protomer movements along ssDNA for mechanical transition between RAD51 single- and double-filament without RAD51 dissociation. This mechanism enables elastic change of RAD51 filament length during structural transitions between ATP- and ADP-states.

Homologous recombination (HR) mediated by ATP-dependent DNA recombinases, such as prokaryotic RecA and eukaryotic RAD51, plays an essential role in the maintenance of genome stability by repairing double-stranded DNA breaks (DSBs) and related lesions[1]. When a DSB occurs, the broken DNA strand is resected, yielding a 3′ single-stranded DNA (ssDNA) overhang, which allows the assembly of recombinase filament[2]. The polymerization of RAD51 recombinases in the presence of ATP is entwined around the newly generated ssDNA, forming an active helical nucleoprotein filament, known as the presynaptic complex[3–5]. This presynaptic complex searches for the homologous double-stranded DNA (dsDNA) and then invades the duplex to form a heteroduplex D-loop. This species then serves as a primer for the DNA polymerase to restore genetic information using the homologous DNA as the template. Thus, recombinases are central to the HR process and are highly conserved in prokaryotes and eukaryotes.

Both prokaryotic and eukaryotic recombinases share a conserved ATPase domain for ATP binding. ATP is required to form an active RAD51 nucleoprotein filament and each protomer of the filament engages a nucleotide cluster[6–10]. The bound nucleic acid triplets retain a B-form-like conformation but are separated by ~8 Å from the adjacent triplets[3,5,11], which results in a ~1.5-fold extension in the length compared to the B-form dsDNA and is crucial to DNA strand exchange. ATP is bound between two protomers. ATP hydrolysis is triggered upon DNA binding by RAD51 but is not required for RAD51-mediated DNA strand exchange. ATP hydrolysis of the whole RAD51 filament does not occur simultaneously and is dominated at the terminal end[12]. Although it is not required for strand exchange, ATP hydrolysis is involved in the filament disassembly, which is required before repair-mediated DNA synthesis by polymerase. Single-molecule imaging and biochemical studies have revealed that dismantling RAD51 from the filament

[1]Institute of Biological Chemistry, Academia Sinica, 11529 Taipei, Taiwan. [2]Institute of Biochemical Sciences, National Taiwan University, 10617 Taipei, Taiwan. [3]Institute of Statistical Science, Academia Sinica, 11529 Taipei, Taiwan. ✉e-mail: joeho@gate.sinica.edu.tw

involves two steps[13–15]. In the fast first phase, ATP hydrolysis leads to the conversion of the extended ATP-bound RAD51–DNA filament (RAD51–ATP filament) to the compressed ADP-bound RAD51–DNA filament (RAD51–ADP filament) without dissociation of the protein from the DNA[15,16]. This conversion is reversible, allowing the inactive compressed filament to revert to the active extended form when ATP is resupplied. Pull-down experiments showed that once RAD51–ATP filament forms, it requires 2% of SDS or 500 mM NaCl to remove RAD51 protomers from bound DNA[15]. More importantly, the previous single-molecule experiment, using a continuous flow setup to wash away any RAD51 dissociated from DNA, clearly demonstrated that RAD51 protomers remain bound to DNA during ATP/ADP structure transition[15]. In the slow second phase, the release of ADP from RAD51 promoters leads to the disassembly of RAD51–ADP filament[16]. In addition, RAD51 protomers can only dissociate from the filament ends and dissociation in vivo often requires protein cofactors such as RAD54, RecQ, and Srs2[16–19].

Near-atomic-resolution structures relevant to this process have only been reported for ATP-bound recombinase filaments. *E. coli* RecA–ATP filament and human RAD51(hRAD51)–ATP filament exhibit extended structures with a helical pitch of 94 Å and 100 Å, respectively, and bound ssDNA extensions of ~50% relative to B-form dsDNA[3,4,20]. Low-resolution EM studies have revealed another type of RAD51 filament with a compressed helical pitch of ~70–75 Å[21,22]. Due to lack of near-atomic-level structural information, the structural transition between ATP-bound and ADP-bound states at the molecular level, and the mechanisms by which protein cofactors regulate this transition and disassembly of RAD51, remain poorly understood.

We report here the cryo-EM structure of the hRAD51–ADP filament that, surprisingly, contains a RAD51 double-filament wrapping around the ssDNA (or dsDNA). Structural analyses and time-resolved cryo-EM studies reveal a stepwise collapsing mechanism for the transition between the ATP/ADP-bound filament intermediates, involving two four-promoter movements along the ssDNA.

## Results

### hRAD51-ADP double-filament structure at 3.1 angstrom resolution

The cryo-EM micrograph and 2D class average images of the hRAD51–ADP filament (Fig. 1a and b) reveals a highly ordered and compressed structure, which is different from the typical elongated hRAD51–ATP (or AMPPNP) filaments (Supplementary Fig. 1). ADP-bound hRAD51 can be assembled with dsDNA or ssDNA to form nucleoprotein filaments. This work focused on ssDNA bound hRAD51–ADP filament because ssDNA filaments assemble better under cryo-EM experimental conditions (Supplementary Figs. 2 and 3). Two-dimensional (2D) averaging of thousands of hRAD51-ADP filament images enhanced the signal-to-noise ratio, confirming that the structure is more compressed and wider than ATP-bound filaments (Supplementary Fig. 1). Subsequent helical reconstruction, by applying a helical rise of 19.6 Å and a helical twist of 48.7°, resolved the cryo-EM map to a resolution of 3.7 Å (Supplementary Fig. 2). The refined helical rise and twist shows a helical pitch of 145 Å and 7.4 protomers per helical turn with a diameter of 108 Å (Fig. 1c). In comparison, the reported hRAD51–ATP filament has a helical pitch of ~100 Å and 6.3 protomers per helical turn, with a diameter of 98 Å (Supplementary Fig. 1)[3]. Closer inspection showed, surprisingly, that the hRAD51–ADP structure forms a two-start helix, containing a hRAD51–ADP double-filament wrapping around ssDNA, unlike the conventional hRAD51–ATP single-filament (Supplementary Fig. 1). Two-start C1 symmetry was then imposed

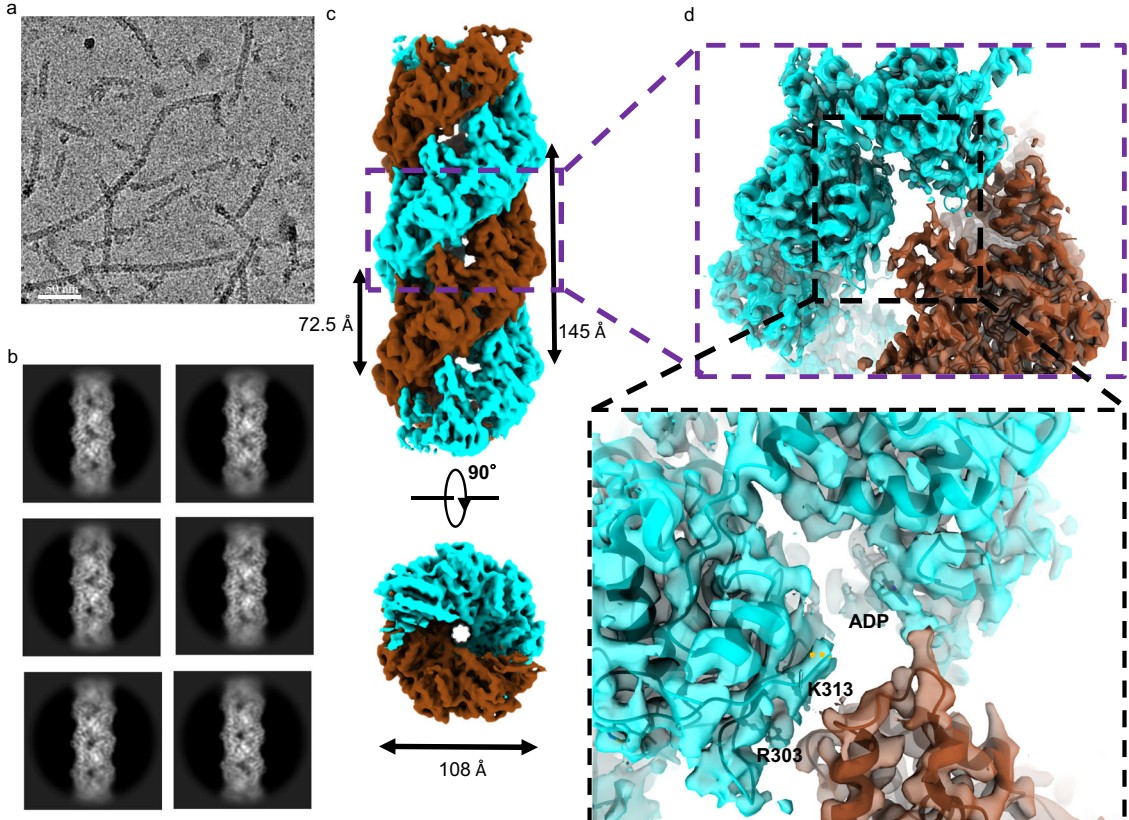

**Fig. 1 | Cryo-EM structure of hRAD51–ADP double-filaments.** A representative (**a**) micrograph and (**b**) 2D class average images of hRAD51–ADP double-filaments are shown. **c** The cryo-EM map of hRAD51–ADP double-filament at 3.1 Å resolution is shown in both side and top views. One filament is colored cyan and the other brown. The helical pitch and diameter are shown. **d** The cryo-EM map focused on filament-filament interaction. The box is the zoom-in view of the interaction region with the side chain of key residues and ADP (shown as sticks) and the protein structure (shown as a ribbon diagram).

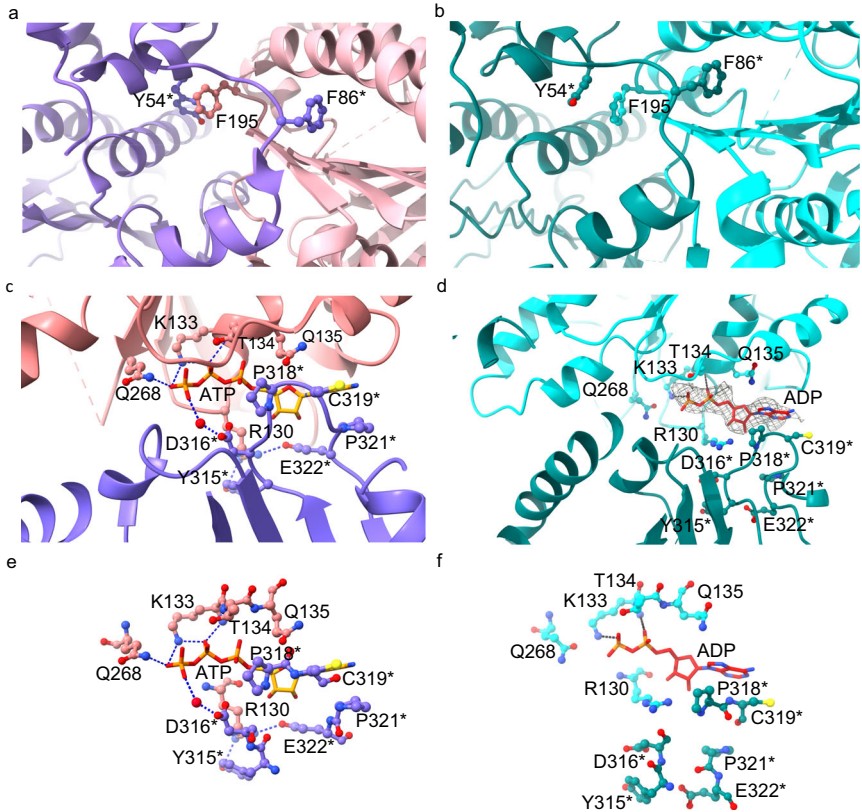

**Fig. 2 | Comparison of hRAD51–ATP and hRAD51–ADP filaments focusing on inter-protomer interaction.** Three inter-protomer interactions in the hRAD51–ATP filament (PDB ID: 5NWL) are shown in (**a**, **c**, and **e**) while corresponding ones in the hRAD51–ADP filament are shown in (**b**, **d**, and **f**). The two protomers in hRAD51–ATP filament are colored pink and purple, respectively; those in hRAD51–ADP filament are colored cyan and green, respectively. Key residues and nucleotides are shown as sticks. **a** and **b** Show Tyr54* and Phe195 aromatic stacking and the Phe86 interdomain linker against the ATPase core. **c** and **d** Show the nucleotide-binding pocket. The cryo-EM map of ADP is shown as a gray mesh. A water molecule, mediating the hydrogen bonds between ATP and D316*, is shown as a red dot. Key residues for the nucleotide interactions are labeled. **e** and **f** Show the detailed ATP or ADP interactions with surrounding residues. The interactions of phosphate with Q268, K133, and T134 are indicated as blue dashed lines.

during helical refinement, after which real-space optimization of the helical parameters yielded final values of 9.8 Å in the helical rise and −155.6° in the helical twist (Supplementary Fig. 2). The axial distance between neighboring strands is 72.5 Å (Fig. 1c). The improvement of cryo-EM map resolution from 3.7 Å to 3.1 Å by applying updated helical parameters and focused refinement not only supported the two-stranded helix but also provided more structural details for functional analyses (Supplementary Fig. 2). Although the individual filament of the hRAD51–ADP double-filament extends the length ~24% compared to the hRAD51–ATP filament (from a helical rise of 15.8 Å to 19.6 Å), it takes two filaments to form hRAD51–ADP double-filament. As a result, the overall filament length is reduced to ~62% (the helical rise changes from 15.8 Å to 9.8 Å).

### Local conformational changes of the nucleotide-binding pocket resulting in extension of the hRAD51 filament

The protomers in ATP-bound and ADP-bound states undergo minimal overall conformational changes, with a root-mean-square deviation of Cα of 0.9 Å, indicating that major conformational changes occur only at inter-protomer interfaces, not within the protomer (Supplementary Fig. 4). The formation of the hRAD51–ATP filament contains three inter-protomer interfaces: (1) Tyr54* and Phe195 aromatic stacking (* indicates from the adjacent protomer), (2) a packing of the Phe86* interdomain linker against the ATPase core, and (3) the nucleotide-binding pocket buried between two protomers[3,11]. While the first two interfaces remain similar (Fig. 2a and b), a major difference occurs in the nucleotide-binding pocket region

(Fig. 2c–f)[3,23]. The conserved Walker A motif (K133, T134, and Q135) is responsible for the binding and hydrolysis of ATP[24], while the loop (Y315*–E322*) from the neighboring protomer sandwiches ATP from the other side (Fig. 2c)[3]. These interactions stabilize the binding of R130 with Y315* and E322*, which connects the two neighboring hRAD51 protomers together for hRAD51–ATP filament formation. The γ-phosphate of ATP interacts with both Q268 on DNA binding loop L2 and D316* on the ATP cap[25], resulting in the buried interface between two protomers (Fig. 2e)[25].

Upon ATP hydrolysis, while the ADP nucleotide is still bound to the Walker A motif, the loss of γ-phosphate eliminates interactions between Q268 and D316* and further destabilizes the R130 interactions with Y315* and E322* (Fig. 2f), resulting in loop Y315*–E322* moving away from the adenosine moiety of the nucleotide. These local changes open up the inter-protomer interface with a rotation angle of 19° and extend the hRAD51 filament (Supplementary movies 1 and 2).

### The ADP-bound state dislodges the L1 and L2 loops from DNA triplet interaction

Each ATP-bound hRAD51 protomer engages one nucleotide triplet of ssDNA and extends the spacing between it and a neighboring triplet as the signature structural feature of RecA/RAD51 nucleoprotein filaments[3-5,20]. Arg235 on Loop 1 (L1) and Val273/Asp274 on Loop 2 (L2) insert themselves into the space between the two triplets and stabilize the inter-triplet base separation, resulting in an extension of the ssDNA or dsDNA to ~1.5 times the length of B-form

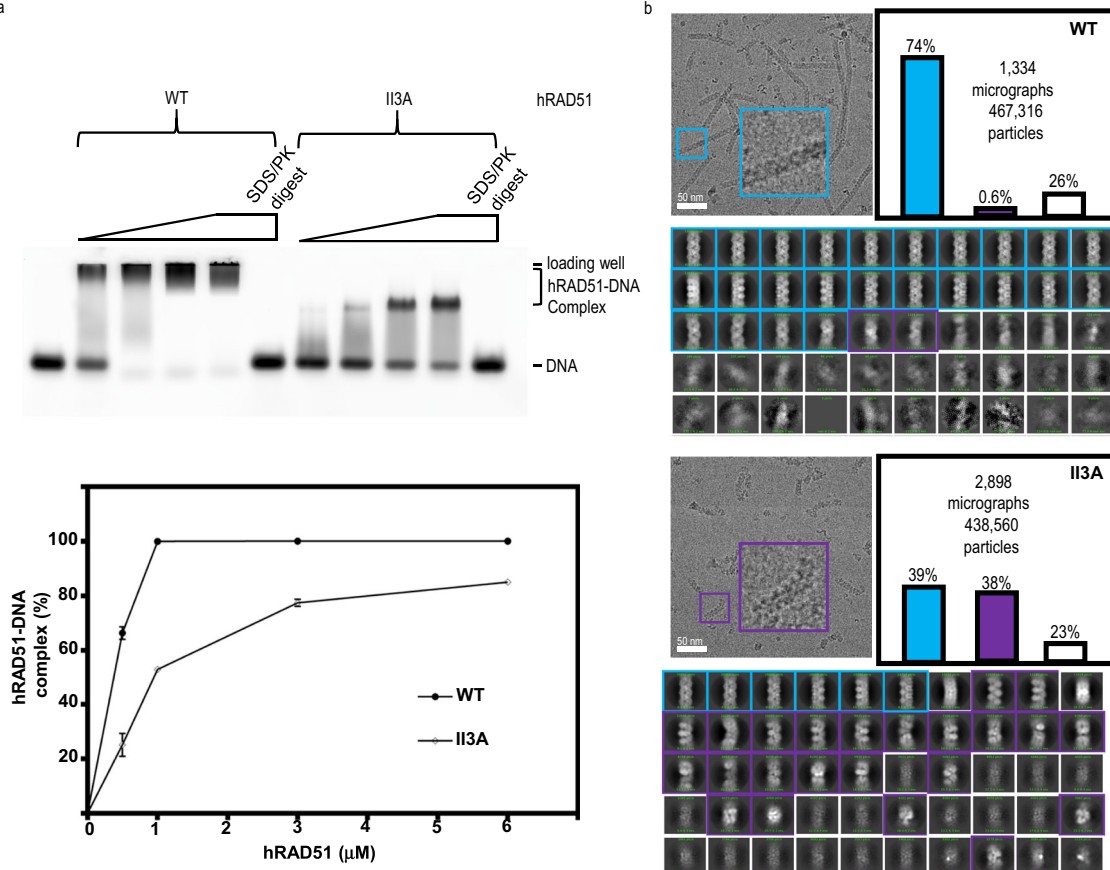

**Fig. 3 | The comparison of hRAD51 wild-type and II3A in the double-filament formation. (a)** EMSA analysis of DNA-binding activities of hRAD51 wild-type and II3A variant. The indicated concentration of hRAD51 was incubated with an 80 nt ssDNA in the presence of 2 mM ADP. In the lane of PK/SDS, the reaction mixture was treated with proteinase K (PK, 0.8 mg/ml) and SDS (0.1%) at 37 °C for 15 min to release the DNA from the nucleoprotein complex as a control experiment. A plot of the % complex versus concentration of RAD51 was graphed with the average values ± s.e.m. from three independent experiments. $N = 3$ **(b)** cryo-EM analyses of the hRAD51–ADP double-filaments are shown. Representative micrographs from hRAD51 wild-type (top) and II3A (bottom) are shown in the upper left corner. Examples of the enlarged filament of ordered hRAD51–ADP and disordered hRAD51–ADP are shown in the cyan and green boxes, respectively. Fifty classification images from all particles are shown and are categorized into three types: ordered hRAD51–ADP (cyan boxes), disordered hRAD51–ADP (purple boxes), and unknown (white boxes). Particle distribution percentages of ordered hRAD51–ADP (cyan bar), disordered hRAD51–ADP (purple bar), and unknown (white bar) are shown as bar graphs. The number of micrographs and particles for the distribution analyses is indicated.

dsDNA[3,5,26,27] (Supplementary movie 1). The γ-phosphate of ATP plays a critical role in positioning L2 by interacting with Q268[28]. During ATP hydrolysis, the loss of γ-phosphate changes the Q268 conformation and dislodges L2 from the spacing between triplets (Supplementary movies 1 and 2). Moreover, the inter-protomer rotation induced by ATP hydrolysis also moves L1 away from the central axis of the hRAD51 filament and dislodges it from triplet interaction. Taken together, the movement of Loop 1 and Loop 2 in the ADP-bound state dislodges Arg235, Val273, and Asp274 from inter-triplet spacing allowing the extended ssDNA or dsDNA inside the filament to transform back to the original length[5,26]. While the central cavity of the hRAD51–ADP double-filament remains positively charged for DNA binding (Supplementary Fig. 5), the wider cavity (from 98 Å to 108 Å in diameter) and lack of Loop 1 and Loop 2 interaction gives ssDNA greater mobility inside the filament, resulting in weak and diffused ssDNA density that can be observed in the cryo-EM map at a lower contour level (Supplementary Fig. 6). To confirm that ssDNA is required to form hRAD51–ADP double-filament. Our EM studies showed that no hRAD51–ADP double-filament could form in the absence of ssDNA and nanogold particle can be observed aside from the hRAD51–ADP double-filament when ssDNA is chemically labeled with a nanogold particle of 10 nm (Supplementary Fig. 7).

## Filament dynamics between ATP single-filament and ADP double-filament monitored by ATP-chase and time-resolved cryo-EM

The cryo-EM structure of the hRAD51–ADP double-filament suggests that two RAD51 single-filaments collapse into a RAD51 double-filament (Fig. 1c). The N-terminal domain and the site II region form an additional interface for the double-filament (Fig. 1d, supplementary Fig. 8). The site II region, consisting of a cluster of positively-charged residues that relies on the groove of hRAD51–ATP single-filament, is critical for recruiting the homologous dsDNA[29]. Unfortunately, the local cryo-EM map of this additional interface is not sufficient to resolve the specific residue-residue interactions (Supplementary Fig. 9). The RAD51-II3A variant, containing three point mutations in site II (R130A, R303A, and K313A), has been shown to cause defects in HR[29]. Our EMSA experiments showed that hRAD51-II3A proteins, in the presence of ADP, reduce ssDNA binding affinity and form less homogenous nucleoprotein filaments as the complex bands are smeared (Fig. 3a). The cryo-EM experiments further showed that the hRAD51-II3A proteins form shorter and more heterogenous hRAD51–ADP double-filaments (Fig. 3b), confirming the site II region is also important in hRAD51–ADP double-filament formation.

Previous single-molecule imaging studies have shown that the structural transition between ATP-bound and ADP-bound states is

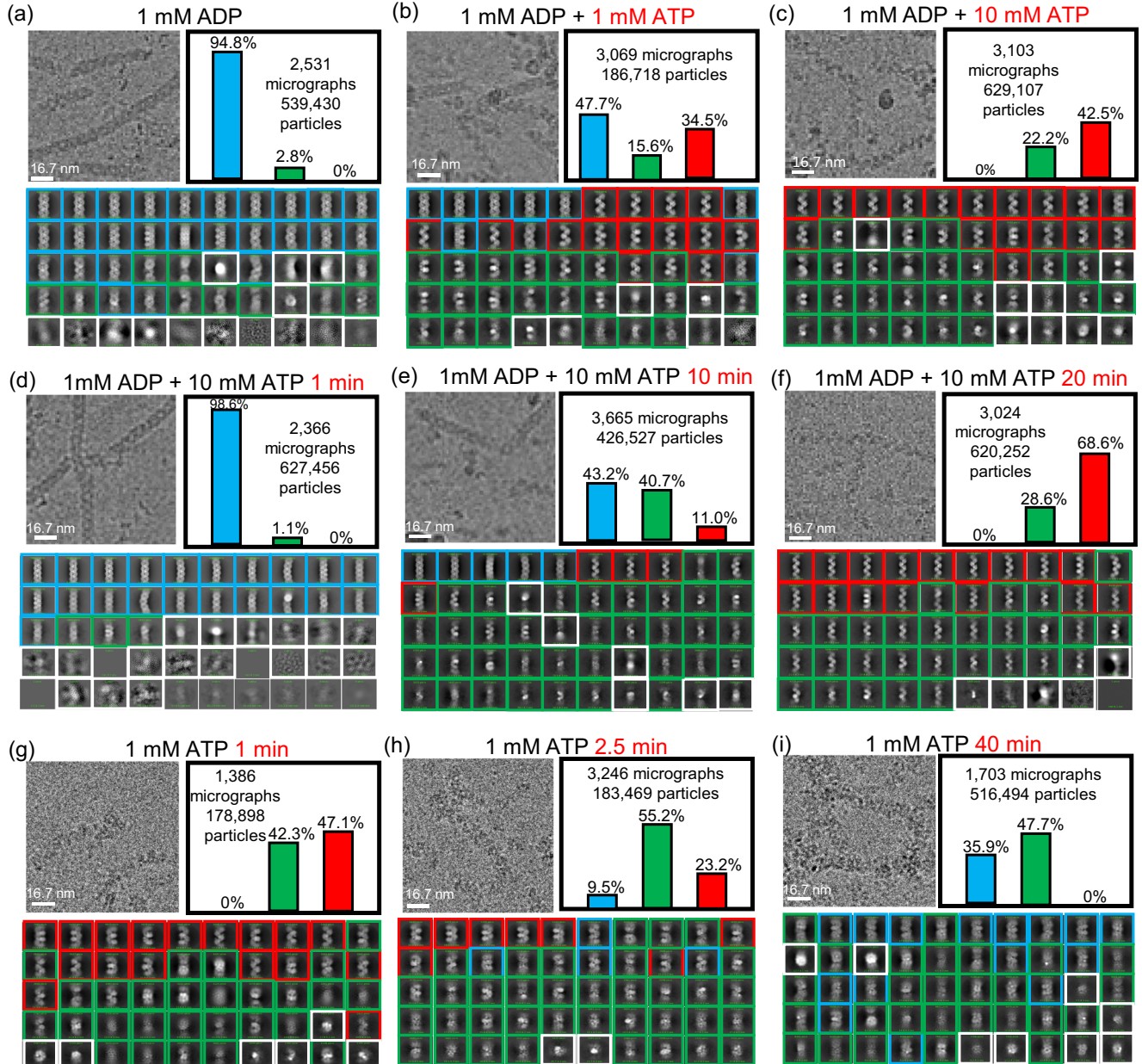

**Fig. 4 | Conformational dynamics of hRAD51 filament monitored by ATP chase and time-resolved cryo-EM. a–c** ATP-chase cryo-EM at different ATP concentrations with incubation times of 15 min and (**d–f**) 10 mM ATP chase with incubation times ranging from 1 min to 20 min (**g–i**) ATP hydrolysis with times ranging from 1 min to 40 min are shown. One representative micrograph from each condition is shown on the top left. Fifty classification images from all particles are shown in the bottom and are categorized into four types: hRAD51–ATP (red boxes), ATP/ADP intermediate (green boxes), hRAD51–ADP (cyan boxes) and unknown (white boxes). Particle distribution percentages of hRAD51–ATP (red bar), ATP/ADP intermediate (green bar), and hRAD51–ADP (cyan bar) are shown as bar graphs. The number of micrographs and particles for the distribution analyses is indicated.

reversible and that ATP hydrolysis causes the reduction of RAD51 nucleoprotein filament length[15]. This study and the other also demonstrated that the extended hRAD51–ATP filament undergoes a structural transition to the compressed hRAD51–ADP filament while hRAD51 remains bound to DNA[15,16]. Our structure suggests that the reduction of filament length upon ATP hydrolysis is due to the formation of a double-filament in the ADP-bound state[15]. To confirm that the hRAD51–ADP double-filament can revert to the hRAD51–ATP single-filament, we monitored hRAD51 filament dynamics by ATP-chase and time-resolved cryo-EM experiments. The hRAD51–ADP double-filaments were formed by pre-incubation of 1 mM of ADP, followed by ATP chase for 15 minutes and vitrification to preserve the structure. Cryo-EM micrographs clearly show that mostly hRAD51–ADP double-filaments are observed when ATP is not supplied. When the ATP concentration increased to 10 mM, the population of hRAD51–ATP single-filaments increased from 0 to 42.5% while the population of the hRAD51–ADP double-filaments decreased from 94.8 to 0% (Fig. 4a–c). We can also observe the population of single-filaments increased and the population of double-filaments decreased in time-dependent manner when 10 mM ATP is supplied (Fig. 4d–f). In addition, our time-resolved cryo-EM experiments also show that the population of hRAD51–ATP single-filaments decreased and the population of hRAD51–ADP double-filaments increased upon ATP hydrolysis (Fig. 4g–i). Taken together, our ATP-chase and time-dependent cryo-EM experiments show that our hRAD51–ADP double-filaments were freely exchangeable with hRAD51–ATP single-filaments, consistent

with previous single-molecule studies[15]. Between ATP- and ADP- states, a well-defined density map of ssDNA can be observed in the hRAD51−ATP single-filament while the diffused and weak DNA density map can be observed in the hRAD51−ADP double-filament, indicating that the ssDNA still remains in the RAD51−ADP double-filament but lacks specific binding.

### Proposed collapsing mechanism for ATP-ADP structural transition

Two previous single-molecule experiments demonstrated that RAD51 remains bound to DNA during the structural transition between ATP- and ADP- states[15,16]. Therefore, the transition between hRAD51−ATP single-filament and hRAD51−ADP double-filament must be mechanistically smooth if, indeed, the double-filament is formed in the single-molecule experiments. In addition, the transition should occur in steps, meaning that while only portions of filament transform into the ADP-bound form, the hRAD51−ADP filament ends should still connect with neighboring hRAD51−ATP filaments.

We found that the individual hRAD51−ADP strand is more stretched than the hRAD51−ATP filament. When the first protomer of the individual hRAD51−ADP filament is aligned with its counterpart in the hRAD51−ATP filament, the fourth protomer (Supplementary Fig. 10) completely detach from the fifth protomer in the hRAD51−ATP filament. This suggests the hRAD51−ADP filament can completely break apart between the fourth and the fifth protomer upon ATP hydrolysis. More importantly, the fifth protomer is now separated by ~180° from the first allowing the remaining filament to move along the axial direction of the ssDNA without crashing into the newly converted hRAD51−ADP filament (Fig. 5 and supplementary movie 3). Once ATP hydrolysis occurs at the fifth to the eighth protomers, the loss of ssDNA binding on dislodging the L1 and L2 loops allows the two four-protomer hRAD51−ADP filaments to slide along the ssDNA. In addition, the retraction of ssDNA provides the hRAD51−ADP filament sliding force and direction. In the final stage, the first and second four-protomer hRAD51−ADP filament and (the first to fourth and the other from fifth to eighth protomer) slide along the ssDNA and intertwine perfectly. Our collapsing mechanism allows stepwise movement because the first and eighth protomers still remain connected to the neighboring hRAD51−ATP filament whose bound ATP is not yet hydrolyzed (Fig. 5 and supplementary movie 3). In addition, this proposed mechanism also allows the reversible to transform from ADP-bound to ATP-bound state (Supplementary movie 4). If the breakdown of the filament occurs at a different protomer position, the sliding movement of hRAD51 along the axial direction of ssDNA would be structurally hindered unless hRAD51 protomers dissociate from the ssDNA (Supplementary movie 5). Our proposed collapsing mechanism, involving four or eight protomers, agrees with previous observations that 5–10 protomers are involved in each hRAD51 disassembly burst[16] and that four coordinated protomers estimated by Hill coefficient are involved in hRAD51 filament extension in steps[30].

### Supporting evidence for collapsing mechanism

In both ATP-chase and time-resolved cryo-EM experiments, a significant portion of the ATP/ADP-bound intermediates do not resemble hRAD51−ATP or hRAD51−ADP filaments providing additional structural evidence for the proposed collapsing mechanism. The initial 2D class average images of ATP/ADP-bound intermediates showed some with distinct morphological differences compared to hRAD51−ATP and hRAD51−ADP filaments (Fig. 4 and supplementary Fig. 11). Although structural heterogeneity of the ATP/ADP-bound intermediate results in noisy 2D class images to preclude near-atomic-resolution cryo-EM structure determination, low-resolution molecular envelopes at the resolution of ~18 Å could still be reconstructed by the conventional single-particle approach. Our proposed intermediate state models fit well into the low-resolution molecular envelope of ATP/ADP-bound

intermediates, showing the existence of our intermediates in cryo-EM experiments (Supplementary Fig. 12).

In addition, we applied an advanced image denoising approach, RE2DC, to enhance 2D classification for samples possessing structural heterogeneity properties (Supplementary Figs. 11 and 13), enabling us to generate 284 reasonable 2D class average images from 104,466 particles[31,32]. These class average images captured by our cryo-EM experiments should resemble 2D projections of structural intermediates derived from the collapsing mechanism. For that, we created three different ATP/ADP-bound intermediate models and generated a series of 2D projections (50 views) from each as well as ATP and ADP-bound filaments with different tilt angles and axial translations. By systematic comparison of the overall shape and intensity landscape of those generated 2D projections with 2D class average views from our cryo-EM experiments, we could find 53 2D class average images resembling 2D projections of the intermediate steps, providing additional evidence for the proposed collapsing mechanism (Supplementary Fig. 14). Taken together, our low-resolution cryo-EM map and 2D class analyses provide evidence for the proposed ATP/ADP-bound intermediates, supporting our collapsing mechanism.

## Discussion

Decades of studies have shown that the hRAD51−ADP filament contributes to the functional regulation of RAD51. Upon ATP hydrolysis, active hRAD51−ATP filament quickly converts into the disassembly-competent hRAD51−ADP filament[12]. The release of ADP followed by RAD51 protomer dissociation from DNA is slow[33], which allows transformation back to the active form in the presence of ATP[15]. We hypothesize that this tightly-packed hRAD51−ADP double-filament may serve as a safeguard for homologous recombination because the possibility to find only one copy of homologous DNA as the template for DNA repair is low. In our hypothesis, if the homologous recombination process fails to find the complementary strand and to finish strand exchange before the ATP hydrolysis, the homologous recombination process, due to the presence of ADP form, can be reactivated when ATP is resupplied (Fig. 6).

Our cryo-EM work reveals a near-atomic-resolution structure of compressed hRAD51−ADP double-filament, which is structurally different from hRAD51−ATP single-filament. To address filament dynamics between the ATP-bound and ADP-bound states, we propose a three-step collapsing mechanism. In Stage 1, upon ATP hydrolysis, a four-protomer filament extends the filament length ~1.2 fold and rotates outward to break the interaction with the fifth protomer. In Stage 2, ATP hydrolysis occurs in the fifth to eighth protomer, allowing it to slide along the ssDNA. In Stage 3, these two four-protomer filaments intertwine to form a double-filament while both ends of the double-filament are still attached to the adjacent ATP-state filament (Fig. 5 and supplementary movie 3). Our model not only allows the reversible contraction–elongation motion observed in RAD51 filament dynamics but also the filament length change is coordinated with the length change from extended dsDNA to B-from dsDNA[30,34].

The previous single-molecule experiment demonstrated that hRAD51 disassembly can only occur from the ADP-bound state and hRAD51 protomers dissociate exclusively from filament ends[16]. Our double-filament structure showed that DNA is densely coated with ADP-bound RAD51 but the DNA remains mobile due to lack of specific interactions, allowing RAD51−ADP double-filament to slide along DNA. The highly packed protomers in ADP-bound RAD51 provide structural explanations of why protomers can only be dissociated from the filament ends. In addition, the tightly packed hRAD51−ADP double-filament explains the requirement for a translocase or helicase, such as RAD54, Srs2, or RecQ to accelerate dissociation from filament end[17–19,35]. We hypothesized that the requirement of translocase from the heteroduplex DNA might serve as a checkpoint to make sure the

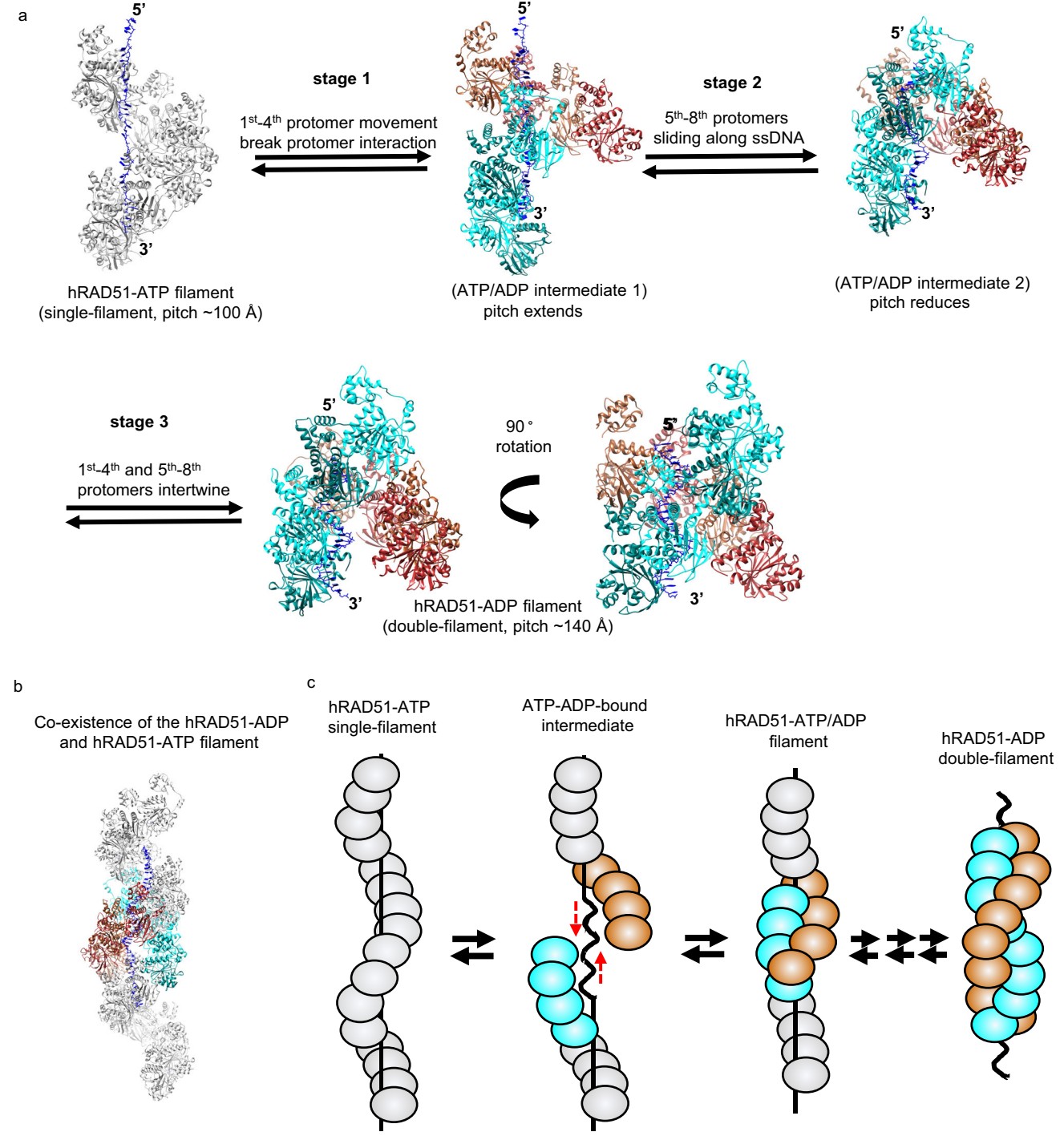

**Fig. 5 | Proposed collapsing mechanism of structure transition between hRAD51–ATP and hRAD51–ADP filaments. a** The proposed collapsing mechanism. Eight protomers are shown as ribbon diagrams. ATP-bound form is colored gray. During ATP hydrolysis, the first four and last four are colored cyan and brown, respectively. The bound ssDNA is shown as blue sticks and 5'/3' ends are indicated. Stage 1: Upon ATP hydrolysis of the first four protomers, this four-protomer filament extends and breaks the interaction with the fifth protomer. Stage 2: Upon ATP hydrolysis of the last four protomers, L1 and L2 no longer interact with the extended ssDNA so it can retract back to the original length. The ssDNA retraction

provides direction and force to help the four-protomer filament on the top to slide down. Stage 3: Two four-protomer filaments intertwine to form a hRAD51-ADP double-filament. The structural transition is reversible. **b** Both ends of the hRAD51–ADP filament (cyan and brown color) still remain connected with hRAD51–ATP (gray color). **c** Cartoon model of the proposed mechanism. RAD51–ATP protomers are colored in gray, whereas RAD51–ADP protomers are colored in brown and cyan. The black straight and curved lines represent extended DNA and original DNA, respectively.

strand exchange is completed before hRAD51 promoters are released from the damaged DNA site (Fig. 6).

Finally, the protein cofactor, HOP2-MND1, possesses a leucine zipper that fits into the helical groove of the hRAD51–ATP single-

filament[36–38] and keeps the RAD51 filament active, even when ATP is hydrolyzed[37]. Our proposed collapsing mechanism illustrates the change of the helical groove between the active ATP state and inactive ADP state, suggesting that the HOP2-MND1 binding to the helical

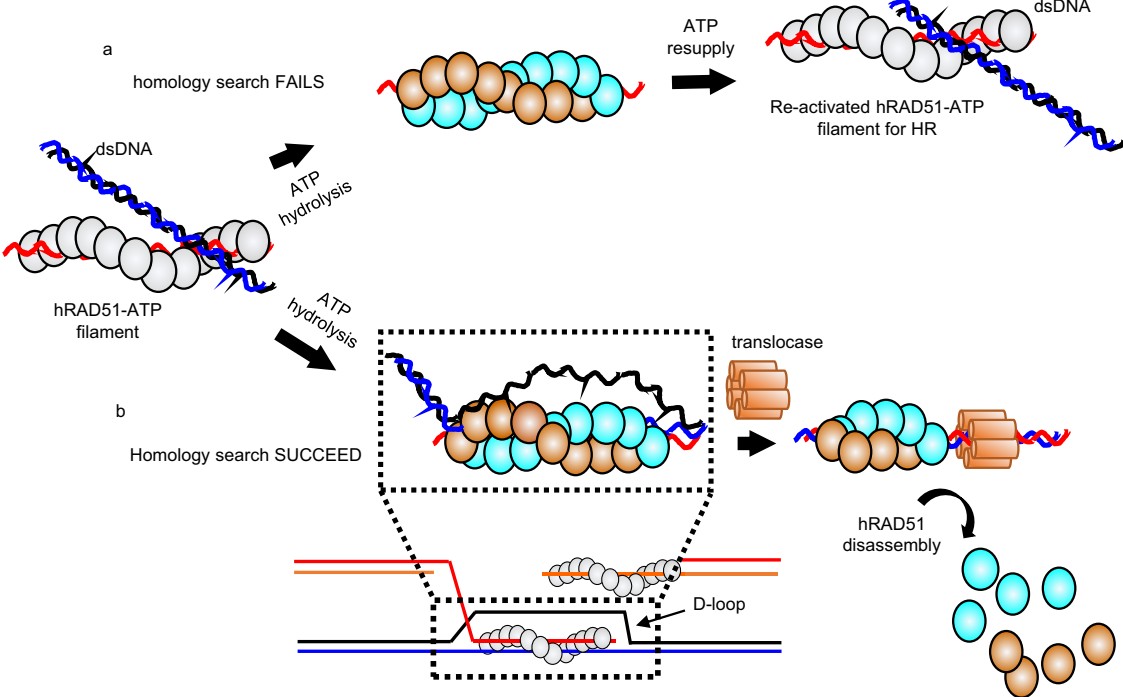

**Fig. 6 | Proposed function of the hRAD51–ADP filaments in HR.** The cartoon model is illustrated as same as Fig. 5c, but ssDNA, homologous ssDNA and displaced ssDNA are drawn as red, blue, and black curved lines, respectively. **a** Due to the presence of ADP form, the HR process can be re-activated when ATP is re-supplied. **b** The requirement of translocase from the heteroduplex DNA to dissemble hRAD51–ADP filament serves as a checkpoint to make sure that strand exchange is completed.

groove provides steric hindrance to prevent structural transition upon ATP hydrolysis, and to keep hRAD51 filament active. Although small molecules inhibiting hRAD51 have been developed for cancer treatment[39–42], this study provides a mechanism for the allosteric control of hRAD51 through the assembly of double-filament, suggesting alternative opportunities for inhibitor design.

Despite our atomic model providing a structural mechanism of filament dynamics reported by previous single-molecule studies[15,16], whether our hRAD51–ADP double-filament represents the compressed form observed in those experiments requires further validation. In particular, the presence of calcium ions can greatly affect hRAD51 filament stability and activities[33,43], and compressed hRAD51–ADP single-filaments in the absence of calcium ions were suggested by low-resolution negatively-stained EM work[22]. Therefore, whether our hRAD51–ADP double-filament is biologically relevant requires further studies since a higher concentration of calcium ions is used in our cryo-EM structure determination.

## Methods
### Protein expression and purification

Human RAD51 protein was expressed in the *E. coli* RecA-deficient BLR strain harboring pRARE plasmid to supply tRNA for rare codons. The recombinant hRAD51 protein was produced and purified as previously described[5,38]. In brief, the cell lysate was subjected to ammonium sulfate (40% saturation) precipitation and resuspended in buffer A containing 20 mM $K_2HPO_4$ at pH 7.5, 0.5 mM EDTA, 10% glycerol, 0.01% Igepal, and 1 mM 2-mercaptoethanol. The following inhibitors were used to prevent protease digestion: PMSF, Benzamidine, Aprotinin, Chymostatin, Leupeptin, and Pepstatin A. The hRAD51 suspension was then purified by Sepharose Q column (GE Healthcare) with a gradient of 150–660 mM KCl in buffer A. The hRAD51 fractions were pooled together and diluted with buffer A followed by the purification by macrohydroxyapatite column (GE Healthcare) using a linear gradient

of 70–560 mM $KH_2PO_4$ in buffer A containing 50 mM KCl. The hRAD51-containing fraction was diluted and further purified by the Source Q column (GE Healthcare) a linear gradient of 235–575 mM KCl in buffer A, and followed by a Mono Q column (GE Healthcare) a linear gradient of 235–490 mM KCl in buffer A. Finally, the RAD51-containing fractions were pooled, concentrated, and aliquoted for storage at –80 °C. The plasmid harboring wild-type human RAD51 cDNA in vector pET11 (Novagen) was subject to site-directed mutagenesis to construct the hRAD51-II3A (R130A/R303A/K313A) mutant. The resulting plasmid was sequenced to ensure no unwanted mutation. The hRAD51-II3A was expressed and purified as described for the wild-type protein.

### Cryo-EM sample preparation and data collection for the hRAD51-ADP filament

The ADP-bound filament assembly reaction was incubated with 6 µM hRAD51 protein and 36 µM 80-nt Oligo 1 (5′TTATGTTCATTTTTTA TATCCTTTACTTTATTTTCTCTGTTTATTCATTTACTTATTTTGTATTA TCCTTATCTTATTTA-3′) in a buffer (35 mM Tris-HCl at pH 7.5, 108 mM KCl and 1 mM DTT) containing 1 mM ADP, 5 mM $CaCl_2$, and 2.5 mM $MgCl_2$ at 37 °C for 30 min. Protein samples were applied on a pre-glow-discharged graphene-oxide-coated Quantifoil holey carbon grid. The grids were blotted and plunge-frozen in liquid ethane using a Vitrobot Mark IV system (Thermo Fisher). Cryo-EM data were acquired on a Titan Krios G3 (Thermo Fisher) microscope operated at 300 KeV, equipped with a Quantum K3 Summit direct electron detector (Gatan), with an energy-selecting slit of 18 eV. Automatic data acquisition was carried out at a nominal magnification of 105,000, yielding a pixel size of 0.83 Å. Movies of 50 frames, corresponding to a total dose of 50 $e^- Å^{-2}$, were collected in super-resolution mode at a dose rate of 1 $e^- Å^{-2}$ per frame and internal defocus range of −1 to −2 µm.

### Cryo-EM data process and helical reconstruction
Our cryo-EM data workflow is summarized in Supplementary Fig. 1. The hRAD51 ADP–filament map was reconstructed by helical

reconstruction. 100 filaments were manually selected to generate a 2D template reference for filament picking automatically. 5,475,512 segments were extracted for 2D classification and 2,622,222 segments were selected for helical reconstruction by cryoSPARC[44]. The initial model without applied symmetry was reconstructed at 4.1 Å. The initial helical rise of 19.6 Å and twist of 48.7° were estimated by a symmetry search. The initial helical rise and twist values, as well as the 2-start C1 helical symmetry, were used to perform subsequent helical refinement, producing a map resolution of 3.58 Å. A two-start 3.58 Å map was used for subsequent non-uniform local refinement and focused refinement to obtain the 3.14 Å helical reconstruction map. The final determined helical rise and twist values were 9.8 Å and −155.6°, respectively. An analysis of the local resolution map estimation was conducted in cryoSPARC[44]. The focus-refined maps (3.14 Å) were stitched together with the map from helical refinement (3.58 Å) using the "vop max" command in UCSF Chimera to obtain a composite stitched map for final structure model building and refinement[45].

### Model building and refinement

The initial hRAD51 protomer structure model was based on the cryo-EM structure of hRAD51−ATP filament[3]. ADP was manually appended in COOT[46] and rigid-body docked into the cryo-EM map in UCSF-Chimera[45]. The refined models were validated and manually inspected using PHENIX[47] and COOT[46]. Cryo-EM data collection, refinement and validation statistics are shown in Supplementary table 1.

### Visualization of DNA in the RAD51−ADP double-filaments by the 5′-nanogold particles

The 5′-biotinylated 80 nt ssDNA substrate for gold labeling was synthesized and purified (IDT). The streptavidin-colloidal nanogold (10 nm, Sigma-Aldrich) and biotinylated 80 nt ssDNA were allowed to react for 1 h at room temperature, which generated a 5′-end-labeled with 10 nm nanogold particles coupled to streptavidin. The hRAD51−ADP double-filaments were prepared by incubation of hRAD51 (400 nM) with the 5′-nanogold-labeled ssDNA (2.5 nM) in the presence of 1 mM ADP at room temperature for 20 min. The protein-DNA complexes were applied to glow-discharged carbon-coated grids. The grids were negatively stained with 1% uranyl acetate for 1 min, and images were obtained from the FEI Tecnai G2 F20 TWIN transmission electron microscope.

### Electrophoretic mobility shift assay (EMSA)

The Cy5-labeled 80 nt ssDNA, Oligo1 (3 μM nucleotides), was incubated with 0.5, 1, 3, and 6 μM of hRAD51 wild-type or II3A variant in 10 μl of reaction buffer (35 mM Tris-HCl at pH 7.5, 1 mM DTT, 10 mM MgCl₂, 100 ng/μl BSA, 50 mM KCl, and 2 mM ADP) at 37 °C for 10 min. The reaction mixtures were resolved in 2% agarose gels in TAE buffer (40 mM Tris, 20 mM acetate, and 2 mM EDTA at pH 7.5) at 4 °C. The gels were subjected to phosphorimaging analysis (Amersham™ Typhoon™ Biomolecular Imager) and quantitative analysis (ImageQuant TL).

### Cryo-EM analysis of the hRAD51−ADP double-filament assembly

The ADP double-filament assembly reaction was incubated with 4 μM wild type hRAD51 or II3A protein and 24 μM nucleotides of 80 nt Oligo 1 in a buffer (35 mM Tris-HCl at pH 7.5, 108 mM KCl, and 1 mM DTT) containing 1 mM ADP, 5 mM CaCl₂, and 2.5 mM MgCl₂ at 37 °C for 30 min. Cryo-EM sample preparation is the same as described above but data were acquired on a Talos Arctica (Thermo Fisher) microscope operated at 200 KeV, equipped with a Falcon III detector. Thousands of micrographs were collected for each condition followed by automatic templated-based particle picking and 2D class averaging analysis. The particle distribution percentages of the well-ordered or disordered ADP double-filaments could be quantified based on particle numbers in each 2D class.

### Structure transition monitored by ATP chase and time-resolved cryo-EM

hRAD51−ADP filaments were assembled under the same conditions as described above. The two-stranded hRAD51−ADP filaments were then chased by ATP at different molar concentrations for 15 mins at 22 °C before vitrification. For time-resolved cryoEM experiments, the hRAD51−ADP filament was chased with 10 mM ATP at increasing time points at 22 °C. For the ATP hydrolysis experiments, hRAD51−ATP filaments at 37 °C were assembled in a buffer containing 35 mM Tris-HCl at pH 7.5, 108 mM KCl, 1 mM DTT, 1 mM ATP, 0.2 mM CaCl₂, and 2.5 mM MgCl₂. The hRAD51-ATP filaments were incubated for various time points before freezing. It is worth mentioning that calcium makes hRAD51 filament more rigid, probably, by organizing DNA bases but high concentrations of calcium completely inhibit ATPase activity of hRAD51[33,43]. To obtain optimized conditions for cryoEM observation of ATP hydrolysis events, 0.2 mM CaCl₂,was added. All protein samples were applied to a pre-glow-discharged graphene-oxide-coated Quantifoil holey carbon grid. The grids were blotted and plunge-frozen in liquid ethane using a Vitrobot Mark IV system (Thermo Fisher). Cryo-EM data were acquired on a Talos Arctica (Thermo Fisher) microscope operated at 200 KeV, equipped with a Falcon III direct electron detector (Thermo Fisher). Data acquisition was carried out at a nominal magnification of 120,000, yielding a pixel size of 0.86 Å. Movies of 50 frames, corresponding to a total dose of 50 e⁻Å⁻², were collected in linear mode at a dose rate of 1 e⁻Å⁻² per frame and internal defocus range of −1.5 to −2.5 μm. Thousands of micrographs were collected for each condition followed by automatic templated-based particle picking and 2D class averaging analysis. The population of hRAD51−ADP double-filament, hRAD51−ATP single-filament, and ATP/ADP intermediate state filaments could be quantified based on particle numbers in each class. For the calculation of the low-resolution envelope of the ATP/ADP intermediate states, a total number of 2,271,743 intermediate state particles were selected from the ATP chase experiments in Fig. 4. After 2D class averaging, 104,466 particles of the intermediate states were selected for further analysis by cryoSPRAC[44]. After ab initio reconstruction, heterogeneous refinement, and homogeneous refinement, the low-resolution envelopes were obtained and docked with the models of ATP/ADP intermediate 1 and 2 using UCSF Chimera[45] (see Supplementary Fig. 12). To quantify the changes, thousands of micrographs were collected under different ATP concentrations, followed by automatic template-based particle picking and 2D class averaging analysis. All fifty 2D class averaging images per condition were categorized into four filament sets and labeled hRAD51−ATP, ATP/ADP intermediate, hRAD51−ADP, or unknown filament. The population of each type was quantified based on particle numbers in the selected 2D classes (Fig. 4).

### Enhanced 2D class averaging and projection similarity between observation and proposed models

For 2D class averaging of the intermediate state, particles belonging to hRAD51−ATP and hRAD51−ADP states were removed. To enhance 2D class averaging of the remaining 104,461 particles, RE2DC, a robust and efficient 2D classification algorithm[32], was used to allow effective classification from a limited amount of particle images per class. Based on the outline of 2D class averaging, 53 out of 286 class averages were chosen for the following analyses and the median number of particles was 23.

For comparison between the proposed model and the observed 2D class average, a similarity score for class average i with respect to the model, which is based on pixel intensity and shape relationship j, is calculated as the inner product of $p_i$ and $(q_j + m_j)$. A higher score means greater similarity. All images were rotated to produce a maximum variance of the vector so the filament inside 2D class averaging images are aligned along the Y axis. Prior to comparison, image masking of 2D class averaging images was performed to reduce error contributed by

noisy background. In brief, Opening, a morphological transformation, was applied to denoise 2D class averaging image, followed by Gaussian blurring to increase the image resolution. The OTSU method is used to perform automatic image thresholding and separate pixels into two classes (intensities 0 and 255) based on a global threshold value from the image histograms. Lastly, a Dilation algorithm was used to enlarge the non-zero area to preserve possible filament edges for mask generation. Masks had a defined filament outline and allowed the removal of noisy backgrounds. Because the intermediate filaments should be attached to ATP- or ADP-bound state filament sections, to focus on the intermediate part of the filament, a window of size $90 \times 90$ for image cropping was chosen based on maximum pixel density to produce target segments for similarity comparison.

To compare our proposed models with the 53 observed class averages, a series of simulations with random direction projections were generated based on the hRAD51–ATP, hRAD51–ADP, and three intermediate models (shown in Supplementary Fig. 14). Projection direction can be decomposed into the three Euler angles ($\varphi$, $\theta$, $\psi$). Since the direction of the filament was already aligned to the Y axis in our observed class averaging images, the $\theta$ angle was limited in the range [86.4–90°] during projection generation. A set of 50 projection images was generated for each model. Masks and the cropping windows for each projection image generated from models are the same for the class averages.

Finally, the highest similarity scores between the observed class averages and simulated 2D projections for each model were recorded. The results show that most class averages matched best those from intermediate 1, while matches with intermediate 2 were found but were rare.

### Reporting summary

Further information on research design is available in the Nature Portfolio Reporting Summary linked to this article.

## Data availability

The structure coordinates and cryo-EM maps generated in this study have been deposited in the Protein Data Bank (PDB) and the Electron Microscopy Data Bank (EMDB) under the accession codes 8GYK and EMD-34372 (focused-refined map), and EMD-36040 (composite map), respectively. All additional coordinate and low-resolution cryo-EM maps related to our proposed collapsing mechanism have been deposited in figshare and are accessible at https://doi.org/10.6084/m9.figshare.20920903. Other additional data related to this paper are available upon request from the corresponding author. Source data are provided with this paper.

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

## Acknowledgements

This work was supported by Academia Sinica (AS-TP-112-L01 to M.C.H. and AS-IA-110-M05 to I.P.T.), National Taiwan University (P.C.), National Science and Technology Council (MOST 111–2326-B-002-019 and MOST 111–2311-B-002-006-MY3 to P.C.), and Taiwan Protein Project (Grant number AS-KPQ-105-TPP and AS-KPQ-109-TPP2). The cryo-EM experiments were performed at the Academia Sinica Cryo-EM Center (ASCEM) and the cryo-EM data were processed at the Academia Sinica Grid-computing Center (ASGC). ASCEM is supported by Academia Sinica (Grant number AS-CFII-108-110) and Taiwan Protein Project. ASGC is supported by Academia Sinica.

## Author contributions

H.Y.Y. and P.C. contributed to protein preparation, biochemical analysis, and related experimental designs. M.Y., and S.C.L. performed cryo-EM related experiments. Y.H.L., H.L.S. and I.P.T. performed enhanced 2D class average and comparison. P.C., Y.H.L., I.P.T., S.C.L. and M.C.H. wrote the paper. S.C.L. and M.C.H. designed the experiments. Y.H.L. M.Y., S.C.L. and M.C.H. analyzed data. M.C.H. supervised the work.

## Competing interests

The authors declare no competing interests.
