## [Peer Review File · Nature Communications]

A RAD51–ADP double filament structure unveils the mechanism of filament dynamics in homologous recombinationEditorial Note: Parts of this Peer Review File have been redacted as indicated to remove third-party material where no permission to publish could be obtained.

REVIEWER COMMENTS

Reviewer #1 (Remarks to the Author):

This is a potentially very interesting paper describing a novel structure for Rad51 filaments. However, I had a number of very minor concerns and one very major one. The major concern involves the lack of any evidence for ssDNA in each of the two strands of this double-stranded Rad51 filament. It is proposed that the ssDNA is mobile, and therefore not seen. Alternatively, ssDNA could be absent in each of the two strands. What evidence do the authors have that there really is ssDNA present? In many cases of disorder, weak density exists for the disordered ligand, domain, loop, etc. Have the authors tried to carefully look at very low density thresholds for a possible signal from the ssDNA? If, in fact, ssDNA is not present in these filaments, the entire model presented for the role of these filaments is irrelevant. The "time resolved" cryo-EM studies do not show interconversion between the two states, but do not exclude it either. One could imagine that when ATP is added subunits might dissociate from the ADP form (with no ssDNA bound) and polymerize on the existing ssDNA oligos to form the conventional single-stranded Rad51-DNA filament. The authors need to convince me, and the readers, that this is not possible.

Specific points:

Line 21) "atomic resolution" is used here and a number of other places. However, to the best of my knowledge, no one has ever achieved atomic resolution (~ 1.2 - 1.3 Å, where atoms are resolved) for any recombinase filament. The 3.1 Å resolution in this paper is a great accomplishment, but it is not atomic resolution, it is near-atomic resolution.

Line 49) "addition domain" should be "additional domain"

Lines 72-74) Refs. 20,21 are cited as describing low resolution cryo-EM studies. However, ref. 21 only used negative stain, while 20 used both negative stain and cryo.

Lines 89-90) it is stated that the compressed double stranded filament is consistent with previous findings (and ref. 21 is cited). This is not true at all. Both refs 20 and 21 show clear single stranded filaments in the ADP state. What is shown in this manuscript is quite interesting and novel, but it is completely inconsistent with these previous studies.

Line 107) A helical twist of 204.3° is given. But the convention going back more than 50 years is that when the twist is $> 180^\circ$, it should be given as a negative angle (generating a left-handed 1-start helix). So this twist value should be written as -155.6° .

Line 237) "to prevent atomic-resolution cryo-EM structure determination" should probably be "to preclude near-atomic resolution cryo-EM structure determination".

Lines 246, 248) "2D project views" should be "2D projections"

Lines 280-281) "ssDNA remains mobile due to lack of specific interactions, explaining why protomers can only dissociate from the filament end" seems to me to be a non sequitur as I fail to see why the explanation follows from the mobility of the ssDNA.

Reviewer #2 (Remarks to the Author):

The authors present an interesting structure of a RAD51 double helical filament bound to ADP and presumably 80-mer ssDNA, although the ssDNA is not seen in the maps. The authors also add different concentrations of ATP to the same sample that formed the double filament and then freeze grid samples at various time points and collect cryo-EM data. This results in a mixture of different

states including the normal ATP-bound single filament state, the ADP-bound double filament state, and some intermediate states that may be discrete enough to classify. This part of the analysis gets a little bit confusing (at least to this reviewer) as it is not fully clear how the different discrete states are distributed within and among the filaments. This approach and analysis of the mixed states and possible transition are nonetheless compelling. The observed structural states lead the authors to develop a model in which successive 4-subunit sections of the filament can slide past one another to transform a single filament into a double filament, apparently without dissociation and re-association of the subunits. Although the model is compelling, there is no functional data in the manuscript to support it – it is not clear if/how the proposed structural transformations are related at all to function. All of the data presented in the paper are structural and there is no relation to function. Although the data are in general well presented and clear, there are also some issues with the clarity of the writing in many places and the writing would need significant improvement.

Major points.

1. The quality of the map appears quite a bit lower than what would be expected for a 3.1Å map. There is very little density for side chains. There is an unaccounted for peak between the adenine ring of ADP and the SH group of Cys-31. Large segments of the model, including the loop from an adjacent protomer that contacts the ADP, do not fit into density. The authors do not discuss the resolution differences across the model.
2. It is conceivable that in the minutes after ATP was added to the ADP-bound double filaments, the subunits dissociated, and then re-associated into the ATP-bound filaments (as opposed to the direct transformation that the authors conclude is occurring). The authors do not really address/discuss this possibility.
3. Even if the structural transition model that the authors propose is actually happening, there is no explanation for what the biological role is. If ATP hydrolysis to ADP promotes filament disassembly (one protomer at a time from the end of the filament), then how does assembly of a more tightly packed double filament help that to happen? The structure doesn't really make biological sense in terms of the role of ATP hydrolysis.
4. There is little attention to the new interactions that are formed as the two filaments interact with one another to form the double filament. Presumably these are new contacts that are not present in the single ATP-bound filaments. The authors could in principle introduce mutations into those residues and determine the effect on activity – presumably this would have to be some type of *in vivo* recombination assay. As it stands there is little if any evidence that the double filament is relevant to function. Are the residues at the new inter-filament contact sites conserved? (Lys40-Glu324; Lys64-Tyr301; etc.) Many of these are not in good density anyway. Some of these interactions are not the same along the model for the filament. Was the structure refined with helical restraints?
5. The authors suggest that the ssDNA is collapsing into B-form along the filament axis, even though the DNA isn't visible. However, there is only one ssDNA strand in the filament, so how/why would it form a B-form DNA? That would require two complementary strands. Along these same lines, the absence of visible DNA in the reconstruction is also a concern. One wonders if the DNA is even there in the double filament, which could conceivably have arisen by disassembly of the filament on ssDNA, and re-assembly of a non-DNA filament.

Minor points.

1. Figure 1, legend. It is not an "electron" density map as in x-ray crystallography (it is not just the electrons of the atoms of the sample that interact with the incident electrons, as is the case for x-rays).
2. The authors state that the DNA transforms back to B-form, but yet there is no density for the bound DNA? How do we know it is there??
3. Lines 49-50. The ATPase domain also contains the primary DNA-binding elements. The way it is worded makes it sound like the "additional" domain is the only one that binds DNA. This is a general problem with the manuscript – imprecise language that can make it confusing in places.

4. Lines 164-165. The authors' use of the term "single-stranded RAD51 filament" is confusing. Is it really a "strand" (as in DNA)? Perhaps "RAD51 single filament" would be better.
5. Fig 2. It would be helpful if the figure had a panel with a full filament and boxes boxes to show where the interacting regions are in the full filaments. As it is the reader has little idea where on the filament the interactions are occurring.
6. Figure S5. There are only 4 cyan subunits (not 8 as stated in the legend).
7. If ADP is supposed to promote dis-assembly, then what would be the purpose of re-assembling the protomers into a double-filament?
8. The authors often refer to a "RAD51-DNA filament". It is unclear in some of the different instances if they mean ssDNA, dsDNA, or both (both can be bound).
9. Lines 64-65. "Disassembly of RAD51-DNA filament from the DNA accompanies ADP dissociation" I'm not sure what this means. When ATP is hydrolyzed to ADP, the RAD51 protomers can dissociate (along with ADP) from the filament end. Another example of how the clarity of the writing could be improved.
10. Line 66 "RAD51 protomers can only dissociate from the terminal end of DNA". I think the authors mean from the terminal end of the FILAMENT (not the DNA).

Reviewer #3 (Remarks to the Author):

The authors determined the structure of the ADP-Rad51-ssDNA complex by cryo-EM and proposed filament dynamics between the ATP- and ADP-bound states to understand the mechanism of Rad51-promoted DNA strand exchange.

The experiments were nicely designed and executed. The results are exciting and would help our understanding of the reaction mechanism.

However, it will be desirable to confirm the structure and their model by other methods. For example, the authors could test their model by mutation analysis. They should, at least, discuss their model with the behavior of reported Rad51 mutants. The structure of *Xenopus* Rad51 with ADP (without DNA) was analyzed by small-angle neutron scattering. The authors can examine if their results are compatible with the neutron scattering experiments.

The title may be misleading. The most important observation is a change in the protomer/protomer contact upon ATP hydrolysis.

We would like to thank the Reviewers for their insightful comments on our original submission. Following are our point-by-point responses to the comments of the Reviewers (reviewer 1: page 1; reviewer 2: page 8; reviewer 3: page 17). The changes are highlighted in red in the annotated copy of the revised manuscript (RAD51ADP_trackchanges_for reviewer only.pdf)

Reviewer #1 (Remarks to the Author):

This is a potentially very interesting paper describing a novel structure for Rad51 filaments. However, I had a number of very minor concerns and one very major one. The major concern involves the lack of any evidence for ssDNA in each of the two strands of this double-stranded Rad51 filament. It is proposed that the ssDNA is mobile, and therefore not seen. Alternatively, ssDNA could be absent in each of the two strands.

1-1) What evidence do the authors have that there really is ssDNA present? In many cases of disorder, weak density exists for the disordered ligand, domain, loop, etc. Have the authors tried to carefully look at very low density thresholds for a possible signal from the ssDNA? If, in fact, ssDNA is not present in these filaments, the entire model presented for the role of these filaments is irrelevant.

Response:

We understand the Reviewer's concerns and thank him/her for his/her invaluable and insightful comments. Indeed, the ssDNA density can be observed at low-density thresholds (shown in the revised manuscript as new supplementary figure 6, see below).

To validate the presence of ssDNA, two additional experiments were conducted.

1) No hRAD51-ADP double-filament can be observed in cryo-EM when the ssDNA is absent (shown in the revised manuscript as new supplementary figure 7, see below). Without ssDNA, only ring-shaped hRAD51 (red arrows) or aggregate (white arrows) can be observed.

2) We also attached a nanogold particle on the ssDNA termini. We can see the nanogold particle is attached to the filaments (shown in the revised manuscript as new supplementary figure 7, see below), supporting the presence of ssDNA in the filament.

Other than a lower contour level and those two additional experiments, we can clearly observe the presence of dsDNA inside the dsDNA bound hRAD51-ADP double-filament (shown in the revised manuscript as new supplementary figure 3).

Line 163-172 in the revised manuscript

“While the central cavity of the hRAD51–ADP double-filament remains positively charged for DNA binding (Supplementary figure 5), the wider cavity (from 98 Å to 108 Å in diameter) and lack of Loop 1 and Loop 2 interaction gives ssDNA greater mobility inside the filament, resulting in weak and diffused ssDNA density that can be observed in the cryo-EM map at a lower contour level (Supplementary figure 6). To confirm that

ssDNA is required to form hRAD51–ADP double-filament, our additional EM studies showed that no hRAD51–ADP double-filament could form in the absence of ssDNA and nanogold particles residing by the hRAD51–ADP double-filaments can be observed when ssDNA is chemically labeled with a nanogold particle of 10 nm (Supplementary figure 7).”

The “time-” cryo-EM studies do not show interconversion between the two states, but do not exclude it either. One could imagine that when ATP is added subunits might dissociate from the ADP form (with no ssDNA bound) and polymerize on the existing ssDNA oligos to form the conventional single-stranded Rad51-DNA filament. The authors need to convince me, and the readers, that this is not possible.

Response:

Thanks to the Reviewer for pointing out the other possibility. The key difference between our proposed model and the alternative one is whether RAD51 can dissociate from DNA during the structural transition between ATP- and ADP-bound states.

The best evidence to rule out the possibility of dissociation is the single-molecule experiment carried out by Prof. Eric Greene’s and Prof. Patrick Sung’s groups (published in PNAS 2009, Structural transitions with human RAD51 nucleoprotein filaments. DOI :10.1073/pnas.0811465106). In this reported result, structural transitions of RAD51 filament were directly visualized by the total internal reflection fluorescence microscope (TIRFM) using DNA curtains tagged with a fluorescent quantum dot at one end. Single DNA extension and compression in length upon hRAD51 association, ATP hydrolysis, or ATP re-activation can be monitored in real time (the experimental setup is illustrated below).

[redacted]

In their chase experiment, the additional buffer was added through a flow channel that creates a continuous flow and, ultimately, **wash away any unbound or dissociated hRAD51. This setup clearly demonstrated the human RAD51 undergoes structural transitions upon bound nucleotides without dissociating from DNA** (PNAS figure 1 is shown below, and all other figures of Greene's PNAS publication). Note that RAD51 can be dissociated only at high salt condition (500 mM NaCl; In PNAS figure 2, the presence of 500 mM NaCl prevent any form of RAD51-nucleoprotein filament formation).

Fig 1 of Greene's work : Visualizing the behavior of human Rad51 nucleoprotein filaments.

[redacted]

A shows a schematic of the TIRFM system used with a QD tagged DNA substrate. **B** shows a kymogram of a single DNA molecule and the black line demarks the QD tag at the end of the molecule. **C** shows the same kymogram superimposed with the particle-tracking data used to quantify the length of the DNA over time. The particle-tracking data alone are shown in **D**, and the assembly and chase phases of the reaction are denoted by blue and yellow background, respectively. When Rad51 is injected into the flow cell, assembly of the nucleoprotein filament causes lengthening of the DNA molecule and alters the position of the QD. Chasing the assembled filaments with wash buffer causes them to decrease in length. For the Top and Middle, "T" demarks the tethered end of the DNA molecule, "F" denotes the labeled end of the DNA in the absence of Rad51, and "E" denotes the location of the DNA after assembly of the extended nucleoprotein filament. In the Bottom, the length of the DNA is shown in microns (μm), and time is shown in seconds for all 3 panels.

In addition, another experiment using magnetic beads to capture DNA also validated that hRAD51 remains bound to DNA. After loading an excess amount of hRAD51 onto DNA and washing away excess unbound RAD51 (shown as supernatant in figure 3C of Greene's PNAS work shown below), the RAD51 did not appear in the wash condition but appeared in 2%SDS elution, which is a denatured buffer to elute bound RAD51 from DNA.

In this publication, they demonstrated that both ATP hydrolysis and ATP reactivation processes are reversible while RAD51 remains bound.

Fig 3 of Greene's work : ATP stabilizes the elongated Rad51 filaments. Rad51 filaments were assembled and then chased with buffer that contained varying concentrations of ATP but lacked free Rad51

[redacted]

A shows the effects of 0 mM and 2 mM ATP on human Rad51, and the bar graph in **B** shows the calculated shortening rates for 0, 0.25, 0.5, 1.0, and 2.0 mM ATP. The SDS/PAGE in **C** shows the magnetic bead pull-down assay performed with no ATP during the wash (lanes 1–3), or with 1 mM ATP (lanes 4–6) or 2 mM ATP (lanes 7–9) included in the wash buffers.

Although our cryo-EM experiment cannot completely rule out the possibility of RDA51 dissociation-association mechanism during hRAD51 filament dynamics. Previous single-molecule studies (Greene's work has been cited in the original manuscript) clearly ruled out that possibility. **We are sorry that we did not explain previous findings explicitly in the original version. In the revised manuscript, we emphasize the previous findings to convince readers that RAD51 dissociation-association is not possible.**

Line 64-69 in the revised introduction.

Pull-down experiments showed that once ATP-bound RAD51-DNA filament forms, it requires 2% of SDS or 500 mM NaCl to remove RAD51 protomers from bound DNA. More importantly, the previous single-molecule experiment, using a continuous flow setup to wash away any RAD51 dissociated from DNA, clearly demonstrated that RAD51 protomers remain bound to DNA during ATP/ADP structure transition.

Line 191-198 in the revised result

Previous single-molecule imaging studies have shown that ATP hydrolysis causes the reduction of RAD51 nucleoprotein filament length¹⁵. **This study and the other also demonstrated that the extended hRAD51-ATP filament undergoes a structural transition to the compressed hRAD51-ADP filament while hRAD51 remains bound to DNA^{15,16}.** Our structure suggests that the reduction of filament length upon ATP hydrolysis is due to the formation of a double-filament in the ADP-bound state¹⁵. **The**

same study showed that structural transition between ATP-bound and ADP-bound states is reversible¹⁵.

Specific points:

1-3) Line 21) “atomic resolution” is used here and a number of other places. However, to the best of my knowledge, no one has ever achieved atomic resolution (~1.2-1.3 Å, where atoms are resolved) for any recombinase filament. The 3.1 Å resolution in this paper is a great accomplishment, but it is not atomic resolution, it is near-atomic resolution.

Response:

In the revised manuscript, “atomic resolution” is replaced by “near-atomic resolution.”

1-4) Line 49) “addition domain” should be “additional domain”

Response:

This sentence is removed based on the Reviewer 2’s comment to avoid confusion.

1-5) Lines 72-74) Refs. 20,21 are cited as describing low resolution cryo-EM studies. However, ref. 21 only used negative stain, while 20 used both negative stain and cryo.

Response:

Sorry for not making it accurately.

In the revised manuscript, “low resolution cryo-EM studies” is replaced by “low-resolution EM studies” to include both negative stain- and cryo-EM studies in line 78.

1-6) Lines 89-90) it is stated that the compressed double stranded filament is consistent with previous findings (and ref. 21 is cited). This is not true at all. Both refs 20 and 21 show clear single stranded filaments in the ADP state. What is shown in this manuscript is quite interesting and novel, but it is completely inconsistent with these previous studies.

Response:

Sorry for the confusion.

What we tried to emphasize in the original manuscript is that the 2D class average images from our compressed hRAD51-ADP double-filament are consistent with previous 2D class average images observed by others. However, our near-atomic resolution cryoEM structure determination is able to reveal a novel double-filament whereas the previous low-resolution cryoEM determination showed a single filament. We believe if those previous low-resolution maps were able to push to near-atomic resolution, they would also see the double-filament structure. Thanks to the Reviewer. We now realized that our original sentence can be misled and is removed from the

revised manuscript.

1-6) Line 107) A helical twist of 204.3° is given. But the convention going back more than 50 years is that when the twist is $> 180^\circ$, it should be given as a negative angle (generating a left-handed 1-start helix). So this twist value should be written as -155.6° .

Response:

As the Reviewer suggested, the twist value is changed to -155.6 in line 113 and line 381 as well as the Supplementary figure 2 and table 1.

1-7) Line 237) “to prevent atomic-resolution cryo-EM structure determination” should probably be “to preclude near-atomic resolution cryo-EM structure determination”.

Response:

As the Reviewer suggested, prevent atomic resolution is replaced by preclude near-atomic resolution.

1-8) Lines 246, 248) “2D project views” should be “2D projections”

Response:

As the Reviewer suggested, 2D project views is replaced by 2D projections in line 273, 275, 277 and 278 of the revised manuscript.

1-9) Lines 280-281) “ssDNA remains mobile due to lack of specific interactions, explaining why protomers can only dissociate from the filament end” seems to me to be a non sequitur as I fail to see why the explanation follows from the mobility of the ssDNA.

Response:

We are sorry for the confusion. The whole statement is revised in line 308-314

The previous single-molecule experiment demonstrated that hRAD51 disassembly can only occur from the ADP-bound state and hRAD51 protomers dissociate exclusively from filament ends. Our double-filament structure showed that DNA is densely coated with ADP-bound RAD51 but the DNA remains mobile due to lack of specific interactions, allowing RAD51-ADP double-filament to slide along DNA. The highly packed protomers in ADP-bound RAD51 provide structural explanations of why protomers can only be dissociated from the filament ends.

Reviewer #2 (Remarks to the Author):

The authors present an interesting structure of a RAD51 double helical filament bound to ADP and presumably 80-mer ssDNA, although the ssDNA is not seen in the maps.

2-1) The authors also add different concentrations of ATP to the same sample that formed the double filament and then freeze grid samples at various time points and collect cryo-EM data. This results in a mixture of different states including the normal ATP-bound single filament state, the ADP-bound double filament state, and some intermediate states that may be discrete enough to classify. This part of the analysis gets a little bit confusing (at least to this reviewer) as it is not fully clear how the different discrete states are distributed within and among the filaments. This approach and analysis of the mixed states and possible transition are nonetheless compelling. The observed structural states lead the authors to develop a model in which successive 4-subunit sections of the filament can slide past one another to transform a single filament into a double filament, apparently without dissociation and re-association of the subunits. Although the model is compelling, there is no functional data in the manuscript to support it – it is not clear if/how the proposed structural transformations are related at all to function.

Response:

Thanks for thinking that our approach, analysis, and model are compelling. Our analysis could not distinguish how different discrete states are distributed within and among the filament. This point could be our future research direction to develop a new approach that allows us to trace back each state in the filament.

We did not provide functional data because decades of biochemical and single-molecule works have provided a lot of functional data. As several review articles have pointed out the advance of single-molecule methods in studying homologous recombination. I have listed two examples of review articles: 1. A change of view: homologous recombination at single-molecule resolution. *Nat Rev Genet.* 2018; doi: 10.1038/nrg.2017.92; 2. Mechanics and Single-Molecule Interrogation of DNA Recombination. *Annu Rev Biochem* 2016. doi: 10.1146/annurev-biochem-060614-034352.

By single-molecule approaches, Eric Greene's group has reported the reversible structural transformation between ATP and ADP states (PNAS 2009, doi.org/10.1073/pnas.0811465106) and Gijs Wuite's group has demonstrated that RAD51 protomers are disassembled from the filament ends (Nature 2009, doi.org/10.1038/nature07581). Our structural model provides the structural transformation mechanism at the molecule level to explain those observations by single-molecule approaches.

2-2) All of the data presented in the paper are structural and there is no relation to function. Although the data are in general well presented and clear, there are also some issues with the clarity of the writing in many places and the writing would need significant improvement.

Response:

We provided the relation to function in the revised discussion (also see 2-5 below).

Major points.

2-3) 1. The quality of the map appears quite a bit lower than what would be expected for a 3.1Å map. There is very little density for side chains. There is an unaccounted for peak between the adenine ring of ADP and the SH group of Cys-31. Large segments of the model, including the loop from an adjacent protomer that contacts the ADP, do not fit into density. The authors do not discuss the resolution differences across the model.

Response:

Sorry for the confusion. We are sorry that the map provided previously is the unsharpened map and we are happy to provide a sharpened map for review. In this revised manuscript, we provided the local resolution map of the whole filament (supplementary figure 2) and the local resolution of a RAD51 protomer (supplementary figure 9) to present resolution differences across the model.

We also noticed an extra density between the adenine ring of ADP and the SH group of Cys-31. Originally, we hypothesized that the extra density could be a divalent ion, but, sadly, the local resolution (around 3.8 angstrom) is not sufficient enough to draw a solid conclusion. Since this is not related to our main focus, we decided not to mention it to avoid any confusion.

The density of loop316-322, which is near ADP, can be observed at a lower contour level so those residues have high B-factors in PDB. In the previous cryoEM work of ATP bound hRAD51-ssDNA single-filament by Wang HW group (NSMB 2017, PDB:5H1B and EMD: EMD-9566), the same loop was also built with high B-factor and the corresponding density can be observed at a lower contour level. To avoid any confusion, we decided to remove loop316-322 in our revised PDB (the revised coordinate is re-deposited with the same ID).

B-factor distribution of the loops surrounding the binding nucleotide

Revised PDB with D316-E322 deletion

2-4) 2. It is conceivable that in the minutes after ATP was added to the ADP-bound double filaments, the subunits dissociated, and then re-associated into the ATP-bound filaments (as opposed to the direct transformation that the authors conclude is occurring). The authors do not really address/discuss this possibility.

Response:

Once again, we are sorry about the confusion. This is the same question raised by Reviewer 1. We have addressed this above (page 3-5). In short, the possibility of the dissociation-reassociation mechanism was ruled out by previous single-molecule experiments so our collapsing model provides the molecular mechanism of this structural transformation. We also revised the manuscript to emphasize the previous finding.

Line 64-69 in the revised introduction.

Pull-down experiments showed that once ATP-bound RAD51-DNA filament forms, it requires 2% of SDS or 500 mM NaCl to remove RAD51 protomers from bound DNA. More importantly, the previous single-molecule experiment, using a continuous flow setup to wash away any RAD51 dissociated from DNA, clearly demonstrated that RAD51 protomers remain bound to DNA during ATP/ADP structure transition.

Line 191-198 in the revised result

Previous single-molecule imaging studies have shown that ATP hydrolysis causes the reduction of RAD51 nucleoprotein filament length¹⁵. This study and the other also demonstrated that the extended hRAD51-ATP filament undergoes a structural transition to the compressed hRAD51-ADP filament while hRAD51 remains bound to DNA^{15,16}. Our structure suggests that the reduction of filament length upon ATP hydrolysis is due to the formation of a double-filament in the ADP-bound state¹⁵. The same study showed that structural transition between ATP-bound and ADP-bound states is reversible¹⁵.

2-5) 3. Even if the structural transition model that the authors propose is actually happening, there is no explanation for what the biological role is. If ATP hydrolysis to ADP promotes filament disassembly (one protomer at a time from the end of the filament), then how does assembly of a more tightly packed double filament help that to happen? The structure doesn't really make biological sense in terms of the role of ATP hydrolysis.

Response:

ATP hydrolysis is triggered upon DNA binding by RAD51 but is not required for RAD51-mediated DNA strand exchange. Therefore, the biological function of the ADP-state before filament disassembly has remained a mystery in this field.

It is known that the likelihood for active RAD51 filament to find the only one homologous copy among the entire genome is very slim. In addition, the loading RAD51 onto ssDNA *in vivo* requires an orchestrated effort from protein cofactors, including BRCA2, RAD51 paralog, etc (Genes. 2018 Dec; 9(12): 629. doi: 10.3390/genes9120629). We believe this relatively stable ADP form serves as a safeguard for homologous recombination (HR) to quickly reactivate RAD51 when the HR fails. Without the stable ADP-bound state, the damaged ssDNA might be easily released from RAD51 and then be degraded by endogenous DNA nuclease, resulting in more severe lesions. If the HR succeeds, the translocase from the homologous dsDNA can remove RAD51 from heteroduplex DNA. We have illustrated our proposed mechanism below and briefly discussed this safeguard hypothesis in the revised manuscript (line 289-293).

“This relatively stable RAD51–ADP double-filament may serve as a safeguard for homologous recombination because the possibility to find only one copy of homologous DNA as the template for DNA repair is low. If DNA strand exchange fails, the stable ssDNA bound RAD51–ADP double-filament can be quickly reactivated when ATP is resupplied.”

However, this hypothesis requires further validation and is not the main focus of this study so we decided not to include this illustration in the revised manuscript.

Stable RAD51-ADP as a HR safeguard

2-6) 4. There is little attention to the new interactions that are formed as the two filaments interact with one another to form the double filament. Presumably these are new contacts that are not present in the single ATP-bound filaments. The authors could in principle introduce mutations into those residues and determine the effect on activity – presumably this would have to be some type of in vivo recombination assay. As it stands there is little if any evidence that the double filament is relevant to function. Are the residues at the new inter-filament contact sites conserved? (Lys40-Glu324; Lys64-Tyr301; etc.) Many of these are not in good density anyway. Some of these interactions are not the same along the model for the filament. Was the structure refined with helical restraints?

Response:

We are grateful for the Reviewer's careful analysis the inter-filament interface. The structure is refined with helical restraints. As shown in the revised supplementary figure 9, we don't think the resolution is sufficient to pinpoint any specific interactions as the Reviewer also stated.

However, the new interface is near site II. A triple-mutant on site II region, hRAD51-II3A (R130A, R303A, and K313A) was reported to be defective for HR (Science 2012, DOI: 10.1126/science.1219379). In the revised figure 3, we showed hRAD51-II3A proteins, in the presence of ADP, bind ssDNA weaker and form less homogenous double-filament compared to the wild-type.

line 177-190 in the revised manuscript

“The N-terminal domain and the site II region form an additional interface for the double-filament (Figure 1d and supplementary figure 8). The site II region, consisting of a cluster of positively-charged residues that relies on the groove of hRAD51-ATP single-filament, is critical for recruiting the homologous dsDNA. Unfortunately, the local cryo-EM map of this additional interface is not sufficient to resolve the specific residue-residue interactions (Supplementary figure 9). The RAD51-II3A variant, containing three point mutations in site II (R130A, R303A, and K313A), has been shown to cause defects in HR. Our EMSA experiments showed that hRAD51-II3A proteins, in the presence of ADP, reduce ssDNA binding affinity and form less homogenous nucleoprotein filaments as the complex bands are smeared (Figure 3a). The cryo-EM experiments further showed that the hRAD51-II3A proteins form shorter and more disordered hRAD51-ADP double-filaments (Figure 3b), confirming the site II region is also important in stable hRAD51-ADP double-filament formation.”

2-7) 5. The authors suggest that the ssDNA is collapsing into B-form along the filament axis, even though the DNA isn't visible. However, there is only one ssDNA strand in the

filament, so how/why would it form a B-form DNA? That would require two complimentary strands. Along these same lines, the absence of visible DNA in the reconstruction is also a concern. One wonders if the DNA is even there in the double filament, which could conceivably have arisen by disassembly of the filament on ssDNA, and re-assembly of a non-DNA filament.

Response:

Sorry for the confusion. The B-form dsDNA only applied to dsDNA filament. For ssDNA, we corrected it to the original ssDNA length in the revised manuscript.

Concerns about the absence of DNA and disassembly/re-assembly are also raised by Reviewer 1 so we addressed it above (page 1-5)

2-8) (1. Figure 1, legend. It is not an “electron” density map as in x-ray crystallography (it is not just the electrons of the atoms of the sample that interact with the incident electrons, as is the case for x-rays).

Response:

We replace electron density map with cryo-EM map.

2-9) 2. The authors state that the DNA transforms back to B-form, but yet there is no density for the bound DNA? How do we know it is there??

Response:

Sorry for the confusion. This is the same question raised by the Reviewer 1 and we addressed it above (page 1-2)

2-10) 3. Lines 49-50. The ATPase domain also contains the primary DNA-binding elements. The way it is worded makes it sound like the “additional” domain is the only one that binds DNA. This is a general problem with the manuscript – imprecise language that can make it confusing in places.

Response:

Sorry for the imprecise description. The additional domain description is removed in the revised manuscript to avoid confusion.

line 48 in the revised manuscript

“Both prokaryotic and eukaryotic recombinases share a conserved ATPase domain for ATP binding.”

2-11) 4. Lines 164-165. The authors’ use of the term “single-stranded RAD51 filament” is confusing. Is it really a “strand” (as in DNA)? Perhaps “RAD51 single filament” would be better.

Response:

Thanks for the Reviewer's great suggestion. To avoid confusion, we use RAD51-ATP single-filament as the known ATP filament and RAD51-ADP double-filament as our ADP filament in the revised manuscript.

2-12) 5. Fig 2. It would be helpful if the figure had a panel with a full filament and boxes to show where the interacting regions are in the full filaments. As it is the reader has little idea where on the filament the interactions are occurring.

Response:

A new panel with a full filament and boxes is included in the revised supplementary figures 2 and 12.

2-13) 6. Figure S5. There are only 4 cyan subunits (not 8 as stated in the legend).

Response:

The legend is revised in revised supplementary figure 10.

2-14) 7. If ADP is supposed to promote dis-assembly, then what would be the purpose of re-assembling the protomers into a double-filament?

Response:

As we addressed point 2-5 (page10-11), this could be a safeguard for HR.

2-15) 8. The authors often refer to a "RAD51-DNA filament". It is unclear in some of the different instances if they mean ssDNA, dsDNA, or both (both can be bound).

Response:

Sorry for the confusion. In this case, we mean both. Because RAD51 can form nucleoprotein filaments with either ssDNA or dsDNA.

2-16) 9. Lines 64-65. "Disassembly of RAD51-DNA filament from the DNA accompanies ADP dissociation" I'm not sure what this means. When ATP is hydrolyzed to ADP, the RAD51 protomers can dissociate (along with ADP) from the filament end. Another example of how the clarity of the writing could be improved.

Response:

Sorry for the confusion. The sentence is revised.

Line 70-71 in the revised manuscript.

"the release of ADP from RAD51 promoters leads to disassembly of RAD51-ADP filament"

2-17) 10. Line 66 "RAD51 protomers can only dissociate from the terminal end of

DNA". I think the authors mean from the terminal end of the FILAMENT (not the DNA)

Response:

Thanks to the Reviewer. The terminal end is replaced with the filament ends in the revised manuscript.

Reviewer #3 (Remarks to the Author):

The authors determined the structure of the ADP-Rad51-ssDNA complex by cryo-EM and proposed filament dynamics between the ATP- and ADP-bound states to understand the mechanism of Rad51-promoted DNA strand exchange.

The experiments were nicely designed and executed. The results are exciting and would help our understanding of the reaction mechanism.

3-1) However, it will be desirable to confirm the structure and their model by other methods. For example, the authors could test their model by mutation analysis. They should, at least, discuss their model with the behavior of reported Rad51 mutants. The structure of *Xenopus* Rad51 with ADP (without DNA) was analyzed by small-angle neutron scattering. The authors can examine if their results are compatible with the neutron scattering experiments. (<https://pubs.acs.org/doi/10.1021/bi971000n>)

Response:

Thanks to the Reviewer. In the revised manuscript a hRAD51-II3A triple mutant was tested and we showed that the mutant proteins, in the presence of ADP, reduce ssDNA binding affinity and form less homogenous double-filament formation (revised figure 3).

The Reviewer also brought up a unique study by small-angle neutron scattering. Regretfully, the experimental condition for this neutron scattering experiment did not include RAD51/ADP/DNA for the comparison.

3-2) The title may be misleading. The most important observation is a change in the protomer/protomer contact upon ATP hydrolysis.

Response:

Thanks to the Reviewer. We totally agree that the most important observation is the new additional protomer/protomer contact upon ATP hydrolysis. The "upon ATP hydrolysis" is now included in the title. To distinguish between the reported ATP contacts and the new ADP contacts, we use the "double" filament in the title for the new additional protomer/protomer contact. In addition, previous single-molecule studies often used filament dynamics for structural transition between ATP and ADP states. For that reason, we include "filament dynamics" in the title.

The revised title is A RAD51–ADP double filament structure unveils the mechanism of filament dynamics upon ATP hydrolysis in homologous recombination

REVIEWERS' COMMENTS

Reviewer #1 (Remarks to the Author):

The authors have done a very good job at addressing the points I raised. While many questions still remain about the function of this unusual structure (raised by the other reviewers) the present work should stimulate further studies that will advance the field.

Reviewer #2 (Remarks to the Author):

In this re-submission, the authors have added two main pieces of new data. First, they have added two new lines of evidence to support their claim that the RAD51-ADP-ssDNA double filaments actually contain the ssDNA. This includes new Fig 6b that shows density for ssDNA in a lower contour level of the original map, and new Fig. S7 that shows that the double filaments do not form when ssDNA is not present and include a gold nanoparticle label at the DNA end. This adequately addresses one of the main concerns from the previous reviews. Second, they have added a functional analysis to support the potential relevance of the double filament. Specifically, they have shown that a triple mutant (II3A) at a potential inter-filament contact point that is specific to the double filament has lower ssDNA-binding activity in the presence of ADP (new Fig 3a) and forms less stable double filaments than WT (new Fig 3b). The authors used this II3A mutant because the cryo-EM map was of insufficient quality to clearly show the inter-filament interactions that would be specific to the double filament, which meant that they could not design more targeted mutations. It is not clear why the authors only tested ssDNA binding in the functional assay – an effect on DNA strand exchange, or an in vivo functional defect would have been more relevant (although also complicated by the fact that the site II mutations would also impact binding of incoming dsDNA to the RAD51-ATP-ssDNA presynaptic filament). A defect on ssDNA binding does not inform on the potential relevance of the double filament for strand exchange function.

Overall, the authors have done a reasonable job addressing the three reviewers' concerns. However, I still have significant concerns about the potential functional relevance of the double filament model.

First, if the role of hydrolysis of ATP to ADP is to promote dissociation of RAD51 subunits from the end of the DNA as has long been observed and thought for RAD51 and RecA, then it just doesn't make sense that ADP would promote formation of a more stable double filament structure. The authors suggest that the double filament may serve to protect the 3'-overhang (bound ssDNA) when ATP is depleted, but this is not supported by evidence for this particular cellular context (low ATP). Thus, a role for the double filament during an ATP depletion state is speculative. In fact, forming a double filament would serve to inactivate the initial RAD51-ssDNA filament, as binding of incoming duplex would be blocked. In the Summary (Abstract), the authors refer to their double filament structure as a "disassembly competent ADP-bound RAD51 filament..", but there is no evidence that this state is more competent for disassembly than a compressed ADP-bound single filament (and structurally it would appear to be less competent for disassembly).

Second, to my knowledge, double filaments have not been observed for RecA or Rad51 (or related proteins from this family) before by lower resolution negative stain methods. For RecA, it is well established that the ADP form is a compressed (lower pitch) single filament, not a double filament. The authors mention similar compressed filaments were observed at low resolution for RAD51 (page 3, lines 78-80, ref 21,22-although I could not find an ADP structure in these papers). The authors use a rather specific condition, including 5 mM Ca²⁺, to obtain the double filaments seen by cryo-EM. If the double filaments are truly relevant, they should be observable at low resolution and over a wider range of conditions.

Third, the model that the authors propose for the single filament (ATP) to double filament (ADP

transition involves "sliding" of 4-subunit groups of RAD52 along the DNA and around one another to re-group, which doesn't make reasonable physical sense. Dissociation and re-binding (to form a double filament) seems much more likely. Although the authors present data from low resolution 2D class averages fitted to modeled 4-subunit structures to support their model for the transition, this part of the analysis is low resolution and preliminary.

In summary, although the discovery and structural analysis of double filaments of RAD51-ADP-ssDNA is interesting, the potential functional relevance has still not been established. Non-functional oligomers have commonly been observed for this group of recombination proteins, and it is not yet clear that the double-filament structure is relevant to function.

Minor points

Figure 2. Please show discussed side chains in panels a,b (Phe86*). Label side chains in c,d.

Interaction of gamma-phosphate with E316* is distant (not shown in H-bonds in Fig. 2c). This is one of the main things that could possibly transmit the loss of the gamma phosphate (I.e. ATP hydrolysis) to changes at the inter-subunit interface. However, the interaction is distant in the first place.

Figure S6. "The one stand of hRAD51 filament is colored cyan and the other brown". I don't think it is appropriate to call the single RAD51 filaments "strand" (as in ssDNA). (it is also mis-spelled as "stand" instead of "strand").

Page 6, lines 195-196, "our structure suggests that the reduction of filament length upon ATP hydrolysis is due to the formation of a double filament in the ADP-bound state." (ref 15). It is also possible that the reduction in length in this single molecule experiment is simply due to the standard reduction in pitch to form the compressed (single) helical filament that has been observed with ADP at low resolution for RAD51 (and RecA).

Page 6, lines 221-224, "Two previous single molecule experiments demonstrated that RAD51 remains bound to DNA during the structural transition between ATP- and ADP-bound states (15,16). Therefore, the transition between hRAD51-ATP single filament and hRAD51-ADP double filament must be mechanistically smooth". The authors are assuming that the ADP filaments in these experiments are also double filaments. Yet, ADP-bound single RAD51 compressed filaments have also been observed (by low resolution EM).

Page 7 line 253 "four coordinated protomers are involved in hRAD51 filament extension in steps" (ref 30). This reference inferred the steps of four from a Hill coefficient. The Hill coefficient is often highly over-interpreted.

Fig 5. What is the force that drives ssDNA "retraction"?

Page 8 line 289: release of ADP from RAD51 and RAD52 from DNA is "slow" (ref 33). How slow is it relative to the timescale of the preparation of RAD52-ADP-ssDNA samples for cryo-EM?

Reviewer #3 (Remarks to the Author):

The authors observed a novel structure for Rad51 filament by cryo-EM and proposed a new mechanism for Rad51-promoted DNA strand exchange reaction based on this structure and the dynamics between the ATP- and ADP-bound states.

The experiments were nicely designed and executed. The results are exciting.

The authors nicely revised the manuscript considering the reviewers' comments.

REVIEWERS' COMMENTS

Reviewer #1 (Remarks to the Author):

The authors have done a very good job at addressing the points I raised. While many questions still remain about the function of this unusual structure (raised by the other reviewers) the present work should stimulate further studies that will advance the field.

Response:

We want to thank the reviewer for his/her valuable comments to make our manuscript better. Indeed, we hope our work can stimulate further studies in the recombinase field. To facilitate further studies, we boldly added our hypothesis of having the ADP as the HR checkpoint (a new Figure 6) in the revised discussion.

Reviewer #2 (Remarks to the Author):

In this re-submission, the authors have added two main pieces of new data. First, they have added two new lines of evidence to support their claim that the RAD51-ADP-ssDNA double filaments actually contain the ssDNA. This includes new Fig 6b that shows density for ssDNA in a lower contour level of the original map, and new Fig. S7 that shows that the double filaments do not form when ssDNA is not present and include a gold nanoparticle label at the DNA end. This adequately addresses one of the main concerns from the previous reviews. Second, they have added a functional analysis to support the potential relevance of the double filament. Specifically, they have shown that a triple mutant (II3A) at a potential inter-filament contact point that is specific to the double filament has lower ssDNA-binding activity in the presence of ADP (new Fig 3a) and forms less stable double filaments than WT (new Fig 3b). The authors used this II3A mutant because the cryo-EM map was of insufficient quality to clearly show the inter-filament interactions that would be specific to the double filament, which meant that they could not design more targeted mutations. It is not clear why the authors only tested ssDNA binding in the functional assay – an effect on DNA strand exchange, or an in vivo functional defect would have been more relevant (although also complicated by the fact that the site II mutations would also impact binding of incoming dsDNA to the RAD51-ATP-ssDNA presynaptic filament). A defect on ssDNA binding does not inform on the potential relevance of the double filament for strand exchange function.

Response:

We want to thank the reviewer for acknowledgment of our efforts in addressing the presence of DNA in the filament. This work focuses on the formation of ADP bound double-filament RAD51 nucleoprotein and filament dynamics between ATP-bound

single-filament and ADP-bound double-filament. Therefore, our binding assays and corresponding cryo-EM analysis showed that II3A reduces DNA binding and forms less ordered double-filament, supporting that II3A region plays a role in the ADP-bound double-filament formation. Since ADP-bound double-filament formation upon ATP hydrolysis occurs after DNA exchange and inactive ADP-filament should not have any exchange activity, the potential relevance of the RAD51-ADP double-filament for strand exchange function assay is not in the scope of this current study. Moreover, testing the *in vivo* function of an inactive form could be very complicated and, again, is not the scope of this study. However, as Reviewer 1 points out, our findings can stimulate further studies that will advance the field.

Overall, the authors have done a reasonable job addressing the three reviewers' concerns. However, I still have significant concerns about the potential functional relevance of the double filament model.

First, if the role of hydrolysis of ATP to ADP is to promote dissociation of RAD51 subunits from the end of the DNA as has long been observed and thought for RAD51 and RecA, then it just doesn't make sense that ADP would promote formation of a more stable double filament structure. The authors suggest that the double filament may serve to protect the 3'-overhang (bound ssDNA) when ATP is depleted, but this is not supported by evidence for this particular cellular context (low ATP). Thus, a role for the double filament during an ATP depletion state is speculative. In fact, forming a double filament would serve to inactivate the initial RAD51-ssDNA filament, as binding of incoming duplex would be blocked. In the Summary (Abstract), the authors refer to their double filament structure as a "disassembly competent ADP-bound RAD51 filament..", but there is no evidence that this state is more competent for disassembly than a compressed ADP-bound single filament (and structurally it would appear to be less competent for disassembly).

Response:

Thanks for Reviewer's insights.

Several single-molecule experiments have shown that ADP form is a relatively stable form and disassemble from the filament end (EMBO J. 2018 Apr 3;37(7):e98162. ; Nature. 2009 Feb 5; 457(7230): 745–748). However, we can not 100% certain if our RAD51-ADP double-filament presents the compressed ADP forms observed in single-molecule experiments or not. To avoid any conflict, we removed "disassembly competent" from the summary.

Due to the lack of compressed ADP-bound "single-filament" structure at the near-

atomic (or atomic) resolution, there is no way to compare which our double-filament or “compressed single-filament” observed by low-resolution EM is more competent for disassembly. One could argue how ADP-bound single-filament could have a 33~40% reduction in filament length of ATP-bound single-filament that was observed in single-molecule experiments (PNAS 106 (31) 12688-12693). Especially, the protomer interface of ATP-bound filament has no additional space to be compressed unless individual protomer can have 40% structural changes in length, which is also unlikely. In our double-filament model, the protomer has minimal structural changes. The massive length reduction is due to the split of a single-filament into a double-filament.

Second, to my knowledge, double filaments have not been observed for RecA or Rad51 (or related proteins from this family) before by lower resolution negative stain methods. For RecA, it is well established that the ADP form is a compressed (lower pitch) single filament, not a double filament. The authors mention similar compressed filaments were observed at low resolution for RAD51 (page 3, lines 78-80, ref 21,22-although I could not find an ADP structure in these papers). The authors use a rather specific condition, including 5 mM Ca²⁺, to obtain the double filaments seen by cryo-EM. If the double filaments are truly relevant, they should be observable at low resolution and over a wider range of conditions.

Response

Thanks for the reviewer’s comment. The ref21 used ATP in the absence of calcium ions. In the absence of calcium, the RAD51 filament hydrolyzes ATP easily. Although there is no RAD51-ADP condition in those references, both low resolution indeed observed two filament pitches, which are 100 Å (elongated form) and ~70-75 Å (compressed form). We are sorry for the confusion and revised it accordingly.

Revised:

Low-resolution EM studies have revealed another type of RAD51 filament with a compressed helical pitch of ~70–75 Å^{21,22}.

We don’t know why double-filament was not reported before. One possibility is that most previous structural work focused on the active form and the experimental conditions are either un-hydrolyzed AMP-PNP or ATP_γS to stabilize filaments. Moreover, they often used dsDNA, not ssDNA.

Initially, we also expected a compressed single-filament and used single-filament helical parameters during helical reconstruction. It was only when we built the atomic model allowing us to trace individual protomers, we realized the formation of double-filament. We then reconstructed cryoEM map using 2-start C1 helical symmetry for the

double-filament helical parameters. Therefore, our point is that it is very difficult to distinguish the compressed single-filament or double-filament in the low-resolution map. However, this is just our speculation since we never know how others did their cryoEM experiments.

As the reviewer points out, divalent ions have a huge effect on filament kinetics and stability. We use 5 mM Ca^{2+} and 2.5 mM Mg^{2+} because this provides the best structural result. But in our time-dependent ATP hydrolysis experiment (figure 3g-i), only 0.2 mM Ca^{2+} and 2.5 mM Mg^{2+} is used (which is a more biologically relevant concentration). In this ATP hydrolysis, 2D class average images also showed double-filament but less ordered. For dsDNA bound hRAD51-ADP double-filament (Supplementary Figure 3), no calcium ions were used to form double-filament (the condition is now included in the figure legend).

However, we do acknowledge that there is a limitation of our study since we cannot validate if the double-filament represents those compressed forms shown in the single-molecule studies due to the presence of calcium ions so we mention this limitation in our revised discussion.

In the revised discussion

“Despite our atomic model providing a structural mechanism of filament dynamics reported by previous single-molecule studies^{15,16}, whether our hRAD51-ADP double-filament represents the compressed form observed in those experiments requires further validation. In particular, the presence of calcium ions can greatly affect hRAD51 filament stability and activities^{33,43}, and compressed hRAD51-ADP single-filaments in the absence of calcium ions were suggested by low-resolution negatively-stained EM work²². Therefore, whether our hRAD51-ADP double-filament is biologically relevant requires further studies since a higher concentration of calcium ions is used in our cryo-EM structure determination.”

Third, the model that the authors propose for the single filament (ATP) to double filament (ADP transition involves “sliding” of 4-subunit groups of RAD52 along the DNA and around one another to re-group, which doesn’t make reasonable physical sense. Dissociation and re-binding (to form a double filament) seems much more likely. Although the authors present data from low resolution 2D class averages fitted to modeled 4-subunit structures to support their model for the transition, this part of the analysis is low resolution and preliminary.

Response:

Without considering cellular content or previously reported studies, I would agree that the dissociation-and-rebinding mechanism is much more likely. However, in the

cell, ssDNA is keen to be degraded so it is well-protected. In fact, after resection, ssDNA is first bound and protected by RPA. Later, the replacement of RPA by RAD51 requires the facilitation by BRAC2 and other proteins. In addition, previous studies showed that the dissociation of RAD51 *in vitro* requires the presence of high salt or SDS (PNAS 2009) and the dissociation *in vivo* often requires the facilitation of translocases (Mol Cell 2010, 39(6): 862-872). Taken together, in the cell, it takes a lot of effort to dissociate RAD51 from DNA and re-load it back. Therefore, the dissociation-and-rebinding mechanism may not be likely once we consider the cellular content.

However, we can not 100% sure if our double-filament represents the compressed form observed in the single-molecule experiment. We specifically mentioned this limitation in our revised discussion (see above).

In summary, although the discovery and structural analysis of double filaments of RAD51-ADP-ssDNA is interesting, the potential functional relevance has still not been established. Non-functional oligomers have commonly been observed for this group of recombination proteins, and it is not yet clear that the double-filament structure is relevant to function.

Response:

We want to thank the reviewer for acknowledging that our work is interesting. We understand that the potential relevance has not been established. However, as Reviewer 1 pointed out, our work should stimulate further studies that will advance the field. To facilitate further studies, we boldly added our hypothesis of having the ADP as the HR checkpoint in the revised discussion.

Minor points

Figure 2. Please show discussed side chains in panels a,b (Phe86*). Label side chains in c,d.

Interaction of gamma-phosphate with E316* is distant (not shown in H-bonds in Fig. 2c). This is one of the main things that could possibly transmit the loss of the gamma phosphate (I.e. ATP hydrolysis) to changes at the inter-subunit interface. However, the interaction is distant in the first place.

Response:

Thanks for the reviewer's suggestion. Figure 2 is revised accordingly (see figure below). Although the interaction of gamma-phosphate with E316* is distant, hydrogen-bond interaction is mediated via a water molecule. As the reviewer stated this may be

one of the main things to change the inter-subunit interface, this water molecule is included in the revised figure 2.

Figure S6. “The one stand of hRAD51 filament is colored cyan and the other brown”. I don’t think it is appropriate to call the single RAD51 filaments “strand” (as in ssDNA). (it is also mis-spelled as “stand” instead of “strand”).

Response:

Thanks for the reviewer’s suggestion. We replace strand as filament now.

Revised one:

“The one filament of hRAD51 double-filament is colored in cyan and the other is colored in brown.”

Page 6, lines 195-196, “our structure suggests that the reduction of filament length upon ATP hydrolysis is due to the formation of a double filament in the ADP-bound state.” (ref 15). It is also possible that the reduction in length in this single molecule experiment

is simply due to the standard reduction in pitch to form the compressed (single) helical filament that has been observed with ADP at low resolution for RAD51 (and RecA).

Response:

We address this in the revised discussion (see above).

Page 6, lines 221-224, “Two previous single molecule experiments demonstrated that RAD51 remains bound to DNA during the structural transition between ATP- and ADP-bound states (15,16). Therefore, the transition between hRAD51-ATP single filament and hRAD51-ADP double filament must be mechanistically smooth”. The authors are assuming that the ADP filaments in these experiments are also double filaments. Yet, ADP-bound single RAD51 compressed filaments have also been observed (by low resolution EM).

Response:

We mentioned the other possibility of compressed single-filament in the revised discussion (see above) and specifically mentioned our assumption in the revised sentence.

Revised one:

Therefore, the transition between hRAD51–ATP single-filament and hRAD51–ADP double-filament must be mechanistically smooth **if, indeed, the double-filament is formed in those single-molecule experiments.**

Page 7 line 253 “four coordinated protomers are involved in hRAD51 filament extension in steps” (ref 30). This reference inferred the steps of four from a Hill coefficient. The Hill coefficient is often highly over-interpreted.

Response

We understand this value is estimated and can be over-interpreted so we mentioned it in the revised version.

Revised:

“four coordinated protomers **estimated by Hill coefficient** are involved in hRAD51 filament extension in steps”

Fig 5. What is the force that drives ssDNA “retraction”?

Response:

In ATP form, the L1 and L2 loops specifically bind to ssDNA by inserting into the space between two adjacent base pairs resulting in stretching ssDNA conformation. In our ADP double-filament form, L1 and L2 loops, due to the conformation changes, no

longer insert into the spacing between two base pairs, which drives ssDNA retraction.

Page 8 line 289: release of ADP from RAD51 and RAD52 from DNA is “slow” (ref 33). How slow is it relative to the timescale of the preparation of RAD52-ADP-ssDNA samples for cryo-EM?

Responses:

Looking at the single-molecule data (figure 5d and figure 2d) done by Robertson RB et al (PNAS 2009, 106 (31) 12688-12693), it is safely said that ADP form remains stable even after 1200 or 600 seconds in the absence of Ca²⁺. We incubated our filament for 30 mins in the presence of Ca²⁺ before plunge-freezing.

Reviewer #3 (Remarks to the Author):

The authors observed a novel structure for Rad51 filament by cryo-EM and proposed a new mechanism for Rad51-promoted DNA strand exchange reaction based on this structure and the dynamics between the ATP- and ADP-bound states.

The experiments were nicely designed and executed. The results are exciting.

The authors nicely revised the manuscript considering the reviewers' comments.

Response:

We want to thank the reviewer for his/her efforts in reviewing our work.